# The *"M* and *P"* Technique for Damage Identification in Reinforced Concrete Bridges

**Athanasios Bakalis** [1,*] , **Triantafyllos Makarios** [1] and **Vassilis Lekidis** [2]

1   Institute of Structural Analysis and Dynamics of Structures, School of Civil Engineering,
    Aristotle University of Thessaloniki, GR-54124 Thessaloniki, Greece; makariostr@civil.auth.gr
2   Institute of Engineering Seismology and Earthquake Engineering (ITSAK-OASP), Dasiliou Street, Pylaia,
    GR-55535 Thessaloniki, Greece; lekidis@itsak.gr
*   Correspondence: abakalis@civil.auth.gr

**Abstract:** The seismic damage in reinforced concrete bridges is identified in this study using the *"M* and *P"* hybrid technique initially developed for planar frames, where *M* signifies "Monitoring" and *P* denotes "Pushover analysis". The proposed methodology involves a series of pushover and instantaneous modal analyses with a progressively increasing target deck displacement along the longitudinal direction of the bridge. From the results of these analyses, the diagram of the instantaneous eigenfrequency of the bridge, ranging from the health state to near collapse, is plotted against the inelastic seismic deck displacement. By pre-determining the eigenfrequency of an existing bridge along its longitudinal direction through "monitoring and frequency identification", the target deck displacement corresponding to the damage state can directly be found from this diagram. Subsequently, the damage can be identified by examining the results of the pushover analysis at the step where the target deck displacement is indicated. The effectiveness of this proposed technique is evaluated in the context of straight multiple span bridges with unequal pier heights, illustrated through an example of a four-span bridge. The findings demonstrate that the damage potential in bridge piers can be successfully identified by combining the results of a monitoring process and pushover analysis.

**Keywords:** reinforced concrete bridges; damage identification; instantaneous eigenfrequency diagram; pushover capacity curve; seismic target deck displacement; bridge plastic mechanism

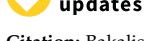



## 1. Introduction

Bridges play a crucial role in infrastructure, emphasizing the need for health monitoring processes to extend their lifespan and ensure safety in the face of environmental factors like earthquakes. Detecting damage in reinforced concrete (RC) structures is a key aspect of these processes, with a primary focus on monitoring alterations in dynamic characteristics to assess the health status. Beyond damage identification, this process plays a pivotal role in establishing dependable structural models. These models, in turn, serve as a foundation for conducting advanced nonlinear analyses, providing insights into the intrinsic seismic capacity of bridges.

The identification of eigenfrequency (and mode shape) along the longitudinal axis of an existing reinforced concrete (RC) bridge can be achieved through instrumental monitoring. This involves the installation of a local multichannel network system of accelerometers to gauge acceleration forces in the longitudinal direction. Notably, fiber optic sensors have gained prominence in recent years for measuring various parameters, including natural frequencies, accelerations, stresses, and strains. The subsequent analytical processing of the recorded response necessitates the application of various stochastic and deterministic procedures. Several techniques developed in the past can be employed for this purpose, such as: (a) the "frequency domain decomposition" technique, which is used in "operational modal

analysis" [1–4]; details on ambient vibration monitoring can be found in the book authored by Wenzel and Pichler [5]; (b) the "stochastic sub-space identification" techniques in which the measured responses directly fit to the parametric models; details can be found in the books authored by Overschee and De Moor [6]. Three distinct algorithms are employed in stochastic subspace techniques: principal component, canonical variate analysis algorithms, and the unweighted principal component; in all cases, random data analysis and operational modal analysis constitute the primary fields of investigation for the recorded accelerograms [7–9]; (c) the "modal time-histories method" [10], which is based on the aforementioned techniques; this method is well suited for structures exposed to earthquake ground excitation or structures experiencing significant wind pressure. Using the "modal time-histories method", eigenfrequencies, mode shapes, and modal damping ratios have been calculated within the linear domain for a variety structures [11]; (d) the "minimum rank perturbation theory" (MRPT), as proposed by Zimmerman and Kaouk [12,13], interprets a non-zero entry in the damage vector as an indicator of the damage location; (e) a technique developed by Domaneschi et al. [14,15], which involves utilizing the discontinuity of mode shape forms; (f) the concept of the damage stiffness matrix is explored in notable works, including those by Peeters [3], Amani et al. [16], and Zhang et al. [17]; (g) techniques that integrate structural health monitoring with pushover analysis are employed for the detection of damage in both individual structural elements [18] and frame structures [19]; (h) several artificial neural network techniques that were developed by Nazari and Baghalian [20] for simple symmetric beams. It is noteworthy to mention the recent research contributions of Reuland et al. [21], who conducted an extensive review of data-driven damage indicators for rapid seismic structural health monitoring. Additionally, Martakis et al. [22] explored the integration of traditional structural health monitoring techniques with innovative machine learning tools, offering a comprehensive perspective. Moreover, [23,24] provide an extensive review of available methods and case studies related to damage identification in bridge structures. Also, details regarding the damage of bearing devices and the abutment–backfill system, along with their impact on bridge behavior, can be found in [25,26]. These works focus on the estimation of seismic fragility curves for common bridges, offering a framework for efficient risk assessment and valuable information on nonlinear modeling techniques and analyses of bridges with various geometrical forms. Some other interesting works offer insights into bridge damage under environmental and operational conditions [27] or to stiffness loss due to heavy vehicle vibrations [28], combining machine learning tools and numerical simulations. In another study [29], the interpretation of dynamic tests on a single-span concrete bridge is presented to explain the observed trend in natural frequencies and is justified by a finite element model analysis.

The current work proposes an alternative and hybrid procedure for identifying damage in existing reinforced concrete (RC) bridges, recognizing that the development of new techniques for this purpose remains an open area of investigation. The technique introduced in this study is based on the "*M* and *P*" technique, where "*M*" stands for "Monitoring" and "*P*" for "Pushover," originally developed by Makarios [30,31] for the identification of damage in planar multistorey RC frames, primarily for seismic loading or wind loading cases.

The primary focus of this study is on damage identification in straight, ductile, RC bridges subjected to seismic loading, specifically along the longitudinal direction of the bridge axis. These bridges feature multiple spans of varying lengths and multiple piers of different heights while the deck spans are simply supported on the various piers. The deck is considered rigid due to the incorporation of continuation plates at the deck level. Adopting the "*M* and *P*" technique for damage identification in RC bridges, along the longitudinal axis, involves the following steps: (i) perform a series of pushover analyses and instantaneous modal analyses in a suitable nonlinear model of the bridge by gradually targeting the Near Collapse state. From the results of these analyses, the eigenfrequency curve (key diagram) is plotted against deck displacement; (ii) utilize instrumental mon-

itoring to identify the fundamental eigenfrequency of the existing (damaged) bridge in the nonlinear regime; (iii) insert the identified fundamental eigenfrequency of the existing bridge into the key diagram. This process reveals the seismic (target) deck displacement corresponding to the monitored eigenfrequency; (iv) from the final step of a pushover analysis targeting the previously found deck displacement, recognize the damage state at the base of the piers. This involves determining the location and severity of damage. Calculate the damage stiffness of the bridge, considering both the developed plastic hinges at the piers and the stiffness degradation of the piers. Additionally, following the final step of the pushover analysis where the deck displacement is attained, perform an instantaneous modal analysis to determine the mode shape of the damaged bridge. This comprehensive procedure provides a systematic approach to identifying and characterizing seismic damage in ductile bridges, considering multiple spans, various pier heights, and the influence of nonlinear behavior.

To validate the effectiveness of the "*M* and *P*" technique for damage identification along the longitudinal direction of RC bridges, an extensive parametric analysis is conducted. This analysis involves the investigation of a group of existing, straight, ductile, multi-span, RC bridges with varying spans and pier heights. A numerical example of an RC bridge with four spans and five piers (each with two columns) of different heights is presented herein, outlining all the steps of the "*M* and *P*" technique in detail. The primary objectives of this study are twofold: (a) calculation of the eigenfrequency curve of the existing (damaged) RC bridge along the longitudinal direction as a function of seismic deck displacement; (b) evaluation of the damage stiffness and the damage image (location and severity of damage) of the RC bridge.

The application of the hybrid "*M* and *P*" technique yields successful results in identifying damage along the longitudinal direction of ductile, RC bridges by combining monitoring methods with pushover analysis. This presents an alternative technique for detecting damage in ductile RC bridges, ensuring accuracy through the "monitoring and identification of frequency". In this study, it is demonstrated that for a given damage pattern in an existing, ductile, RC bridge, the bridge stiffness at the health state undergoes changes. This alteration results in an elongation of the bridge's eigenfrequency, which is experimentally identified through the monitoring procedure. The key diagram of the method is then utilized to determine the seismic deck displacement of the bridge. This displacement corresponds to the identified damage image and ensures the same eigenfrequency value measured in the field. Consequently, the "*M* and *P*" technique stands as a self-evident process, demonstrating its reliability in practical applications.

## 2. Methodology

The focus of this study revolves around a typical scenario featuring an existing straight, multiple-span, reinforced concrete (RC) bridge. The bridge configuration includes piers of varying heights and the deck, which is rigid and is simply supported at the piers. The longitudinal rigidity of the deck is ensured by the presence of continuation plates at the deck level. It is also assumed that the piers exhibit highly ductile behavior, designed in accordance with the High Ductility (DCH) class of EN 1998-1 [32]. Each pier may consist of one or more columns. In an earthquake event, the inelastic behavior of this bridge type along the longitudinal direction is anticipated to manifest at the base of the piers, which exhibit a cantilever behavior. Plastic hinges will develop in this region, leading to damage occurrences at these specific locations.

In order to assess the seismic response of an existing reinforced concrete (RC) bridge with a rigid deck along its longitudinal direction, the bridge can be effectively modeled as a Single Degree of Freedom system (SDOF) [25]. The differential equation of motion for the free vibration of an SDOF, without damping and subject to an initial forced displacement or velocity, is given by:

$$m\,\ddot{u}(t) + k_o\,u(t) = 0 \tag{1}$$

Here, $m$ represents the mass of the bridge, and $k_o$ denotes the stiffness of the SDOF. The variables $u(t)$ and $\ddot{u}(t)$ correspond to the time-varying displacement and acceleration of the SDOF mass, respectively.

If an existing SDOF system presents a damage image during its operational life, attributed to any cause, the stiffness at any time step $i$ will undergo a change by $\Delta k_i$:

$$k_i = k_o - \Delta k_i \tag{2}$$

where $\Delta k_i$ is the change of stiffness due to damage.

Furthermore, the instantaneous mode shape at each inelastic $i$-step of the analysis can be defined by conducting a modal linear analysis using the instantaneous stiffness $k_i$, which incorporates the effects of damage on stiffness. As a result, the equation of motion, Equation (1), is reformulated as follows:

$$m\,\ddot{u}(t) + (k_o - \Delta k_i)\,u(t) = 0 \tag{3}$$

Therefore, given that the mass $m$ remains constant, the eigenvalue problem at the inelastic $i$-step is expressed as:

$$\left[(k_o - \Delta k_i) - \omega^2 m\right]\varphi = 0 \tag{4}$$

Here, $\omega$ (rad/s) represents the instantaneous eigenvalue, and $\varphi$ is the instantaneous mode shape of the SDOF system at the inelastic $i$-step of the analysis. The solution of the eigenvalue problem is obtained by equating the expression within the brackets to zero and determining the $\omega^2$ value from the resulting algebraic equation:

$$\left[(k_o - \Delta k_i) - \omega^2 m\right] = 0 \tag{5}$$

Subsequently, the instantaneous mode shape $\varphi$ can be computed using Equation (4) for the known value of $\omega^2$. Moreover, with the known eigenvalue $\omega^2$, Equation (4) can be reformulated as:

$$\Delta k_i\,\varphi = k_o\,\varphi - \omega^2\,m\,\varphi \tag{6}$$

It is essential to highlight that the identification of the instantaneous frequency $\omega$ and the instantaneous mode shape $\varphi$ of the SDOF structure at the inelastic $i$-step cannot be achieved through the analysis of the records resulted by a time-history analysis (with accelerograms) using random data processing. These procedures necessitate a sufficient time window where the eigenfrequency remains constant, a condition often challenging in seismic scenarios. Instead, the records obtained through a monitoring multichannel network system of accelerometers should originate from the ambient vibration of an existing (damaged) bridge, without any induced motion. This underlines the significance of the ambient vibration data from a stationary (calm) damaged bridge in accurately determining key dynamic parameters for subsequent verification and advanced analytical procedures. Therefore, if $\omega$, $\varphi$, $k_o$, $\Delta k_i$, $m$ are known through the proposed methodology, Equation (6) can be employed at the end for verification purposes (e.g., for advanced optimization and probabilistic analysis).

The identification of damage along the longitudinal direction of multi-span, ductile, RC bridges, with piers of various height, can be achieved through the proposed hybrid "*M* and *P*" technique (where *M* signifies "Monitoring" and *P* denotes "Pushover") that integrates an identification system with a numerical model, according to the following phases:

(a)   The eigenfrequency $f$ of the existing (damaged) RC bridge along the longitudinal direction is identified through monitoring. This involves utilizing a local network of uniaxial accelerometers placed at characteristic positions along the degree of freedom of the system.

(b) A suitable numerical nonlinear model of the RC bridge is established, and a sequence of separate pushover analyses are performed along the longitudinal direction, targeting each time at a gradual increasing deck displacement $u_{deck,i}$. For each target displacement, one pushover analysis is performed leading to the drawing of the capacity curve of the bridge in terms of base shear $V_o$ and deck displacement $u_{deck}$. The base shear of the bridge is computed as the sum of the base shears of the various bridge piers, where each pier can consist of several columns. Figure 1 illustrates the general form of the capacity curve of the bridge along the longitudinal direction. Additionally, the figure features an idealized elasto-perfectly plastic force–displacement relationship, defining the idealized yield point $(u_y, V_{oy})$ of the bridge and the effective (secant) stiffness at the yield point. Various performance levels of the bridge corresponding to different deck displacements $u_i$ are presented in the figure. The Near Collapse state of the bridge is characterized by the ultimate target (deck) displacement, $u_{ult}$. It is noted that P-D effects should always be considered in pushover analysis, especially for more flexible bridge structures. However, caution is advised, as spurious results may arise from the above consideration in the instantaneous modal analyses that follow. This potential issue can be caused from the possible appearance of negative stiffness in the capacity curve due to the P-D effects. Regarding the effective stiffness of the RC bridge piers in the pushover analysis, a proposed stiffness scenario is outlined in the subsequent step.

(c) A sequence of instantaneous modal analysis of the bridge is performed, following the final step of each pushover analysis with an increasing deck (target) displacement $u_{deck,i}$. To be more specific, the stiffness of the existing (damaged) bridge obtained at the final step of each pushover analysis serves as the initial condition for the subsequent instantaneous modal analysis. Utilizing the results from the instantaneous modal analyses, a diagram of the instantaneous (step) cyclic eigenfrequency $f_i$ (in Hz) of the damaged bridge is plotted against the deck displacement $u_{deck,i}$ of the bridge along the longitudinal direction. Figure 2 illustrates the general form of such a diagram, which serves as the key diagram of the "*M* and *P*" technique. In this figure, the inelastic deck displacement $u_{deck,i}$ is represented on the abscissa, while the eigenfrequency $f_i$ of the damaged bridge is on the ordinate. By incorporating the fundamental eigenfrequency $f_1$, identified through the monitoring procedure in phase (a), into this diagram, the seismic inelastic deck displacement $u_{deck,i}$ of an existing damaged RC bridge can be determined.

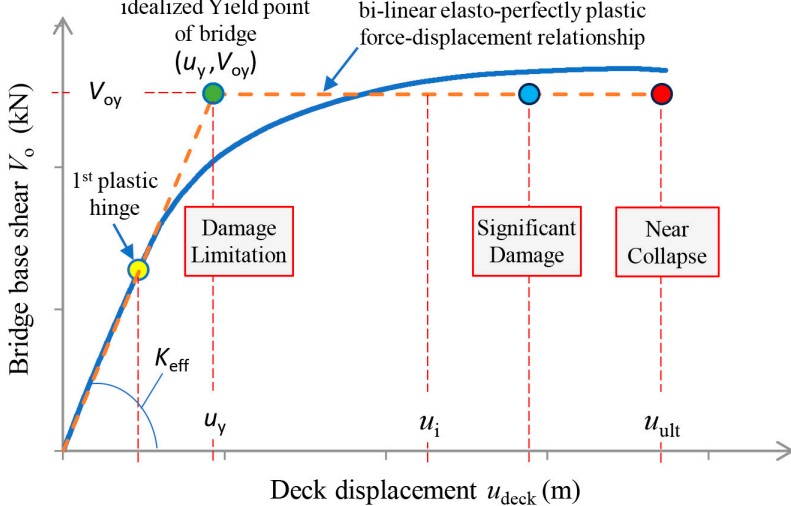

**Figure 1.** Pushover curve of a multispan RC bridge along the longitudinal direction (target displacement at the Near Collapse (NC) state).

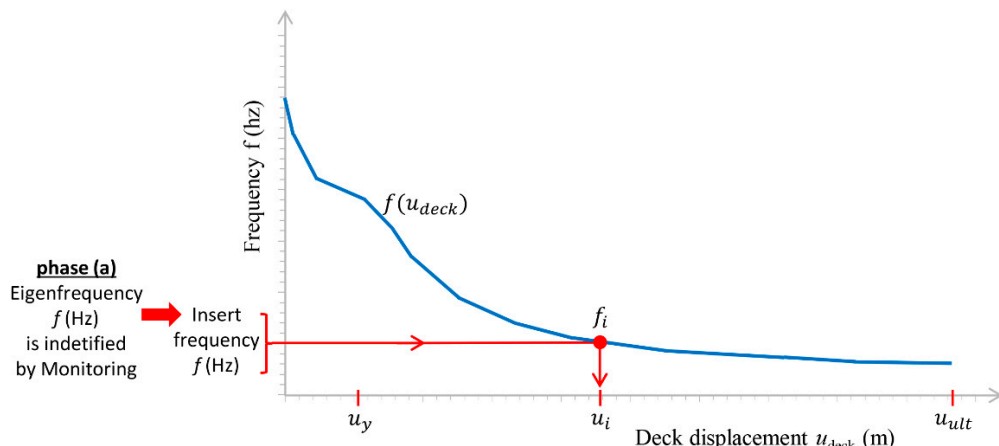

**Figure 2.** Instantaneous eigenfrequency diagram of the RC bridge, in the nonlinear area.

It is important to note that all pushover and instantaneous modal analyses of the reinforced concrete (RC) bridge, targeting each gradual increase in deck displacement $u_{deck,i}$, should be conducted within a suitable nonlinear model of the bridge. This model should account for discrete values $E_c I_{eff,i}$ ($E_c$ is the elastic modulus of concrete) representing the effective bending stiffness of the various piers. This consideration is crucial due to varying levels of stiffness degradation at each target displacement $u_{deck,i}$ corresponding to different damage states (extent and magnitude of cracking along the entire critical plastic region length of the piers). These damage states align with various performance levels, including undamaged (health) state, 1st hinge (1st yield), Damage Limitation (DL), Significant Damage (SD), Near Collapse (NC), and all the intermediate ones (Figure 1).

To address this, an effective stiffness scenario, expressed in terms of the effective moment of inertia ratio $I_{eff,i}/I_g$ of the piers, should be established before performing pushover and modal analyses, as a function of the chord rotation of the bridge piers, $\theta_{c,i}$ (in rad). In this ratio, $I_g$ is the moment of inertia of the geometric sections of the piers. Moreover, the chord rotation of the piers at the examined deck (target) displacement $u_{deck,i}$ is given by $\theta_{c,i} = u_{deck,i}/h_c$, where $h_c$ is the pier height (Figure 3). Consequently, piers of various heights will exhibit different values of chord rotation since the lateral displacement of the rigid deck along the longitudinal direction of the bridge remains the same for all piers at their tops. This approach ensures a comprehensive consideration of stiffness variations associated with different damage states and pier characteristics during the analyses.

The main assumption regarding the effective moment of inertia $I_{eff,i}$ of the RC piers, which exhibit a cantilever bending deformation with potential plastic hinges located at the base-section of the piers, is grounded in the considerations outlined in EN 1998-3 [33]:

$$E_c I_{\text{eff}} = \frac{M_p \cdot L_v}{3 \cdot \theta_y} \tag{7}$$

where $M_p$ is the plastic moment of the base-section of the piers, which is calculated through a section analysis using an elastoplastic idealization of the moment–curvature diagram $M\text{-}\varphi$; $L_v$ signifies the shear span of the piers, which is equal to the height of the piers from the foundation level to the bottom of the deck girders. Additionally, $\theta_y$ denotes the chord rotation of the shear span of the piers at the yield state, approximately given by Equation (A.10) of Eurocode EN 1998-3 [33].

Equation (7) is explicitly derived from elasticity theory, and its rationale is presented in Figure 4. When a cantilever pier undergoes yielding (i.e., when its bending moment reaches the plastic moment $M_p$ of the section), then the effective bending stiffness $E_c I_{\text{eff}}$ of the shear span $L_v$ of the pier ($L_v = M/V$, where $M$ is the bending moment and $V$ is the shear force of the pier) is equal to the secant stiffness of the shear span to the yield point. EN 1998-3 [33] imposes these low values of the secant (effective) bending stiffness at

yield on each pier of the RC bridge in order to perform nonlinear analysis that targets all performance levels, ranging from DL to NC. The adoption of these low values of secant stiffness at yield ensures a conservative approach in displacement calculations, contributing to an overall more flexible behavior of the bridge in the analyses.

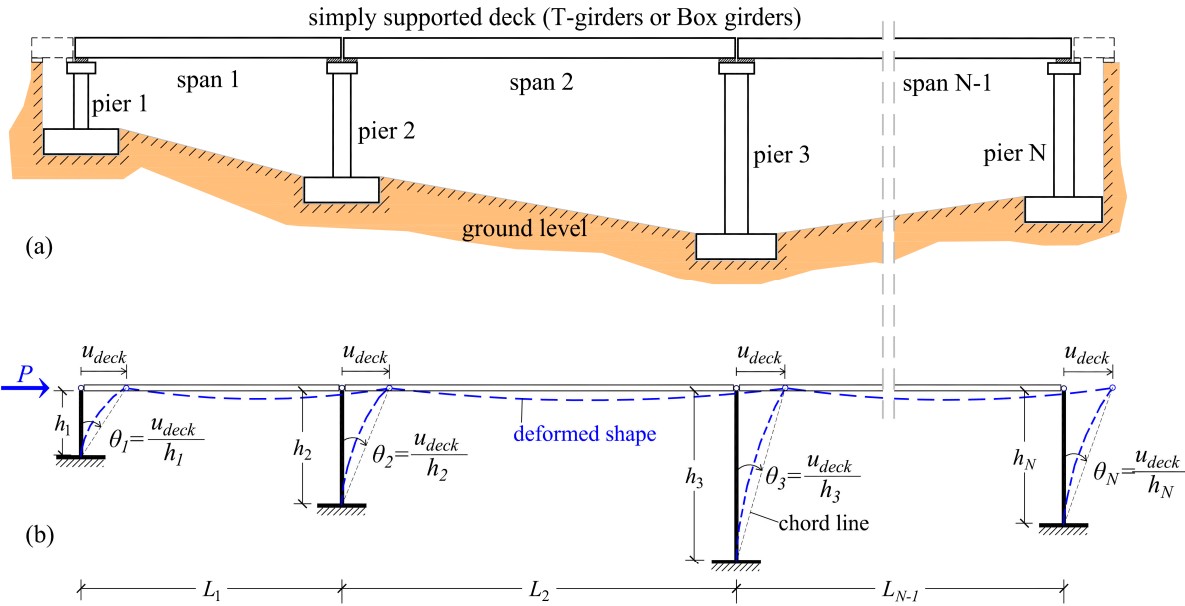

**Figure 3.** (**a**) Simply supported straight bridge with N-1 spans of various lengths and N-piers of various heights; (**b**) chord rotation $\theta_c$ of the bridge piers in pushover analysis with lateral force at the deck level.

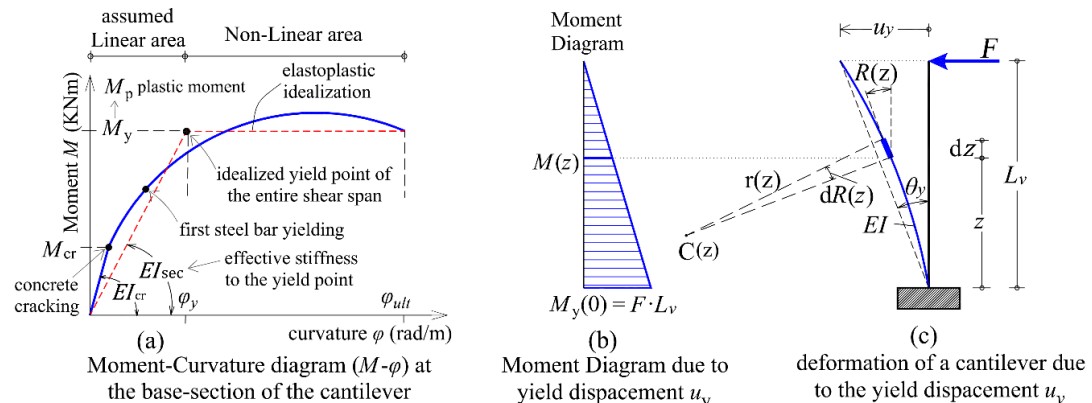

Symbols
$z$: position of examined section, d$z$: finite length, C($z$): curvature center, d$R(z)$: slope variation of bending line, $r(z)$: curvature radius, $\varphi(z)$: curvature at the examined section, $\varphi_y(0)$, $M_y(0)$ and $\theta_y$: yield curvature, yield moment and chord rotation at yield at the base section
Equations: $\varphi(z) = 1 / r(z) = dR(z) / dz$ , $\varphi(z) = M(z) / EI_{sec}$ , $\varphi_y(0) = M_y(0) / EI_{sec}$ , $\theta_y = u_y / L_v = \varphi_y(0) \cdot L_v / 3 = M_y(0) \cdot L_v / (3 \cdot EI_{sec})$

**Figure 4.** Base section analysis of a cantilever pier in the RC bridge, with a height equal to the shear span length $L_v$. This includes the calculation of curvature $\varphi_y$ (rad/m), chord rotation $\theta_y$ (rad), and secant stiffness at yield $EI_{sec}$ (kN·m²) according to EN1998-3 [33].

To mitigate this inherent conservatism in the stiffness scenario depicted in Figure 5, a scaling approach is proposed when the pushover analysis targets higher seismic performance levels, including DL, SD, or any intermediate levels between DL to NC. According to [34], which addresses RC buildings, at the DL state, the suggested value for $E_c I_{eff}$ is equal to $2 \cdot E_c I_{eff,NC}$ and lies between $0.25 \cdot E_c I_g$ and $0.5 \cdot E_c I_g$. At the SD state, $E_c I_{eff}$ is recommended to be equal to the average of the corresponding values at the NC and DL states. Moreover, to simplify the stiffness scenario in order to encompass all piers of the RC bridge, which have various sections, mean values of the ratio of effective stiffness $I_{eff} / I_g$

are proposed as a function of the chord rotation $\theta_{pr}$ (where the subscript "*pr*" denotes profile) for all performance levels. These values are derived from an extended parametric investigation of various ductile RC piers with columns of cyclic cross-sections ($D = 0.6$ to 2 m) and with geometric longitudinal reinforcement ratios ranging from 1% to 4%. This investigation involved consecutive pushover analyses with gradually increasing target displacements, ranging from the health state to the NC state. The analyses incorporated suitable values of effective stiffness determined through a trial-and-error process. The objective was to achieve convergence, aligning the observed chord rotation at the base section of the column piers (performance level) with the assigned percentages of reduction of the moment of inertia. In these analyses, the scaling approach remains within two limits: at the NC state, the effective moment of inertia ratio $I_{eff}/I_g$ of an RC column pier is calculated using Equation (7) of EN 1998-3 [33], while just before DL, at 1st hinge formation (1st yield), the effective moment of inertia ratio is practically equal to the 50% reduction rate proposed by EN 1998-1 [32] for the design of new RC buildings. Similarly, the effective moment of inertia corresponding to achieving the SD performance level must align with the chord rotation at the base of the columns, which, as per EN 1998-3 [33], is approximately 75% of that at the NC level. From the health state to the formation of the 1st hinge, the scaling approach for reducing the effective moment of inertia is constrained within the limits of the geometric moment of inertia $I_g$ and the 50% reduction rate proposed by EN 1998-1 [32], respectively. In the nonlinear model of each cantilever pier, mean values of strengths were employed, and the plastic hinge at the base-section was modeled using the fiber hinge approach with a plastic hinge length in accordance with EN 1998-3 [33]:

$$L_{\mathrm{pl}} = \frac{L_{\mathrm{v}}}{30} + 0.2 {\cdot} h + 0.11 {\cdot} \frac{d_{\mathrm{bL}} {\cdot} f_{\mathrm{ym}}}{\sqrt{f_{\mathrm{cm}}}} \qquad (8)$$

where $f_{\mathrm{cm}}$ is the mean concrete compressive strength, $f_{\mathrm{ym}}$ is the mean yield stress of steel, $d_{\mathrm{bL}}$ is the mean diameter of the tension reinforcement, $h$ is the depth of the cross-section, and $L_{\mathrm{v}}$ is the shear span. This approach ensures a more refined and representative stiffness scenario for the entire bridge, accounting for the variability in pier sections and offering a comprehensive representation of stiffness under different performance levels.

It was found that the mean values of $\theta_{pr,i}$ between all examined RC piers (with different circular sections) at the 1st yield, DL, SD, and NC states were approximately equal to 0.0095, 0.016, 0.068, and 0.091 rad, respectively. The corresponding mean values of $I_{eff,i}/I_g$ were approximately equal to 0.5. 0.4, 0.28, and 0.22. Furthermore, the effective stiffness scenario depicted in Figure 5 introduces discrete $I_{eff,i}/I_g$ values starting from the uncracked (health) state towards the 1st hinge (indicating the 1st yield), and then progressing to DL. At the 1st hinge, the value $I_{eff,i}/I_g = 0.50$ is considered, as suggested in EN 1998-1 [32] for elastic analysis. Additionally, three straight lines for the effective stiffness ratio $I_{eff,i}/I_g$ of the piers within the linear and nonlinear areas are presented in Figure 5. These lines align with the abovementioned proposed $\theta_{pr}$ and $I_{eff,i}/I_g$ values for the 1st yield, DL, SD, and NC states:

For the linear area (health state to 1st hinge), $0 \leq \theta_{pr} \leq 0.00946$ (rad):

$$I_{eff}/I_g = 1 - 52.847 {\cdot} \theta_{pr} \qquad (9)$$

For the assumed linear area (1st hinge to DL), $0.00946 < \theta_{pr} \leq 0.01605$ (rad):

$$I_{eff}/I_g = 0.6436 - 15.174 {\cdot} \theta_{pr} \qquad (10)$$

For the nonlinear area (DL to NC), $0.01605 < \theta_{pr} \leq 0.0913$:

$$I_{eff}/I_g = 0.4384 - 2.391 {\cdot} \theta_{pr} \qquad (11)$$

To utilize this figure, the known chord rotation $\theta_{c,i}$ of the base section of a pier at a discrete target deck displacement $u_{deck,i}$ should be inserted into Figure 5 for $\theta_{pr}$, and the corresponding effective stiffness ratio $I_{eff,i}/I_g$ of the pier can be obtained.

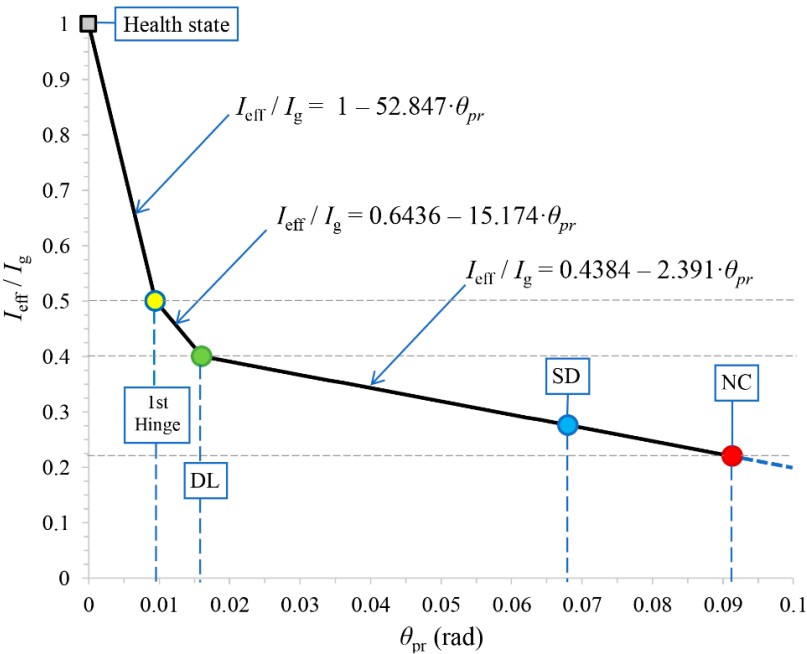

**Figure 5.** Effective moment of inertia ratio $I_{eff}/I_g$ of the ductile piers of the RC bridge at various performance levels (damage states) as a function of the chord rotation $\theta_{pr}$ (rad) at their base section.

(d) The known eigenfrequency $f_i$ (in Hz) of the existing bridge, obtained through the monitoring procedure in phase (a), is incorporated into the instantaneous eigenfrequency diagram (Figure 2). Consequently, the corresponding inelastic seismic (target) deck displacement $u_{deck,i}$ of the bridge is determined from the diagram.

(e) The bridge damage state can be identified through the results of a pushover analysis, specifically targeting the previously determined seismic deck displacement $u_{deck,i}$ from the preceding phase. The location and state of potential plastic hinges that may develop at the base-section of the piers at the last step of pushover analysis provide an estimation of the damaged state of the existing RC bridge. It is important to note that even if no plastic hinge appears at the base of the piers, the magnitude of damage at the base of the piers can still be estimated in terms of stiffness degradation relative to the health state.

(f) Additionally, a linear modal analysis is performed at the final step of pushover analysis in phase (e). The initial conditions for this analysis are derived from the instantaneous stiffness of the RC bridge along the longitudinal direction at this last pushover analysis $i$-step. From the results of this instantaneous modal analysis, both the circular eigenfrequency $\omega_i$ and the mode shape $\varphi_i$ of the damaged bridge are calculated.

(g) Finally, the instantaneous stiffness $k_i$ of the RC bridge along the longitudinal direction is computed at the examined inelastic $i$-step where the deck displacement $u_{deck,i}$ occurs. This calculation is facilitated after determining the flexibility of the damaged bridge at the same i-step. To achieve this, a linear analysis is performed with a lateral force $F_i$ applied at the deck level, specifically along the dynamic degree of freedom of the bridge along the longitudinal direction. This analysis is conducted at the last step of the pushover analysis in phase (e), resulting in the calculation of the corresponding static displacement $u_{st,i}$. Subsequently, the stiffness $k_i$ of the damaged bridge is computed by the ratio $F_i/u_{st,i}$. Thus, the damage stiffness $\Delta k_i$ at the same inelastic $i$-step is calculated using the general equation $\Delta k_i = k_o - k_i$, where

$k_o$ represents the known initial stiffness of the undamaged bridge, determined at the health state as mentioned above. In the preceding linear analysis, the base shear $V_{c,i}$ of the piers is also recorded. Consequently, the stiffness $k_{c,i}$ of the damaged piers can be calculated by the ratio $V_{c,i}/u_{st,i}$, and their stiffness degradation $\Delta k_{c,i} = k_{c,o} - k_{c,i}$ relative to the health state can be determined. Utilizing this information, the location and the magnitude of the damage on the bridge piers can be identified.

In Figure 6, a flowchart illustrating the application of the "*M* and *P*" technique for the identification of damage in RC bridges is presented.

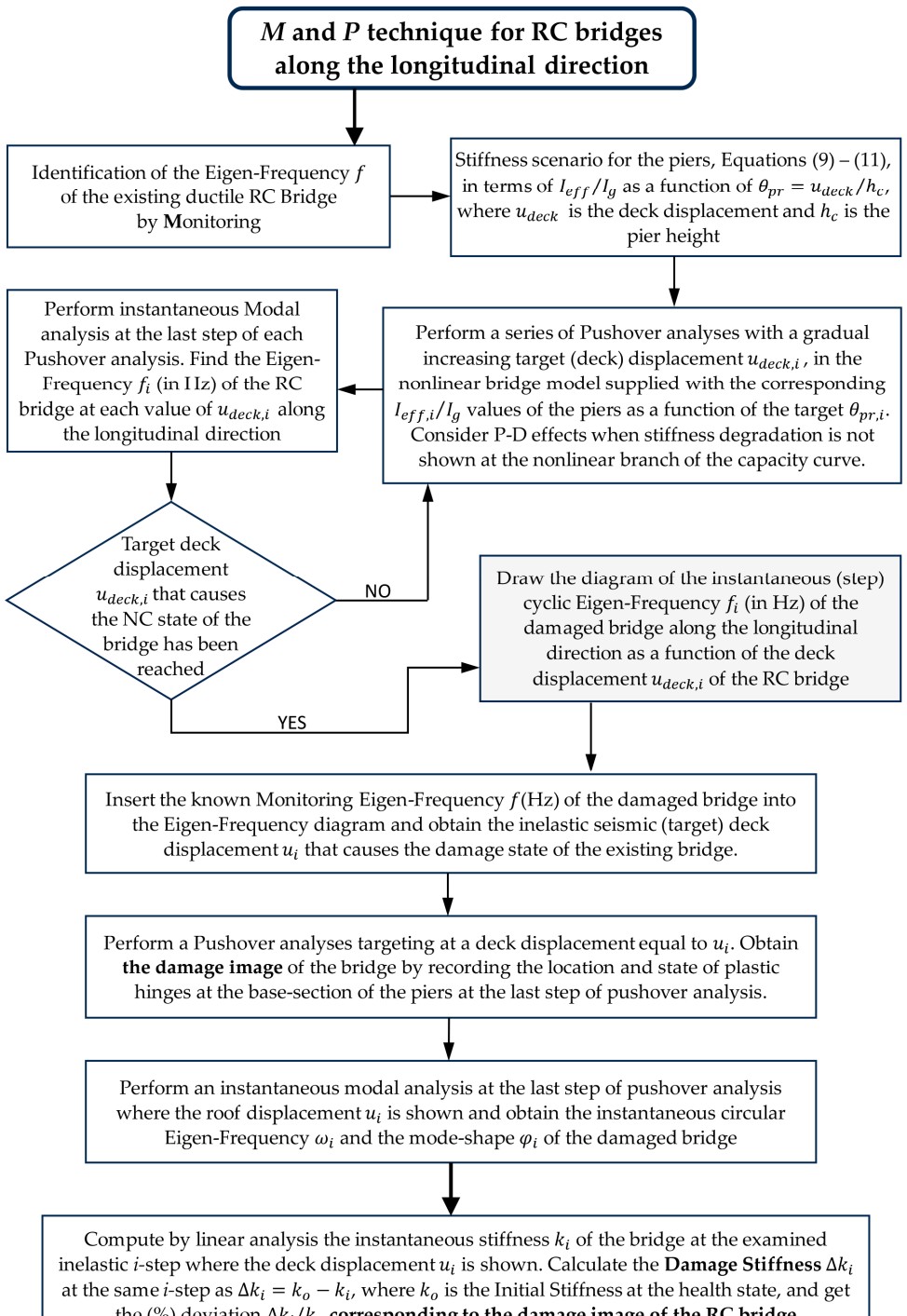

**Figure 6.** Flowchart for the application of the "*M* and *P*" technique in RC bridges.

## 3. Numerical Example

The existing, straight, ductile, RC bridge depicted in Figure 7a comprises four spans with dimensions $L_1 = 15$ m, $L_2 = L_3 = 25$ m, and $L_4 = 20$ m. It features five piers with varying heights: $h_1 = 6$ m, $h_2 = 10$ m, $h_3 = 15$ m, $h_4 = 12$ m, and $h_5 = 8$ m. There are no abutments at the bridge ends due to steep vertical rock slopes in those locations. Each pier consists of two identical circular columns that connect at the top with a cup beam of rectangular section $1.5 \times 1.0$ m. The cap beam extends beyond the piers with a reduced cross-section at the free end. The two circular columns of each pier are assumed fixed into a common slab foundation at the ground level. The column diameters of the five piers are $D_1 = 0.8$ m, $D_2 = 1.2$ m, $D_3 = 1.5$ m, $D_4 = 1.3$ m, and $D_5 = 1$ m. The cast in place bridge deck, with a thickness of 30 cm, is simply supported on the cap beams along each span through precast prestressed girders of a rectangular section $0.5 \times 1.5$ m. The bearings at the top of the cap beams (which are usually of elastomeric type in bridges) are assumed as simple support areas for the girders without altering the dynamic behavior of the bridge. The superstructure of the bridge (deck system) is considered rigid along the longitudinal direction due to the presence of continuation plates above the piers at the deck level. As a result, the dynamic simulation of the RC bridge shown in Figure 7b can be considered as an SDOF system, where the piers exhibit cantilever behavior. A detailed section of the bridge at the location of pier 5 is illustrated in Figure 8a.

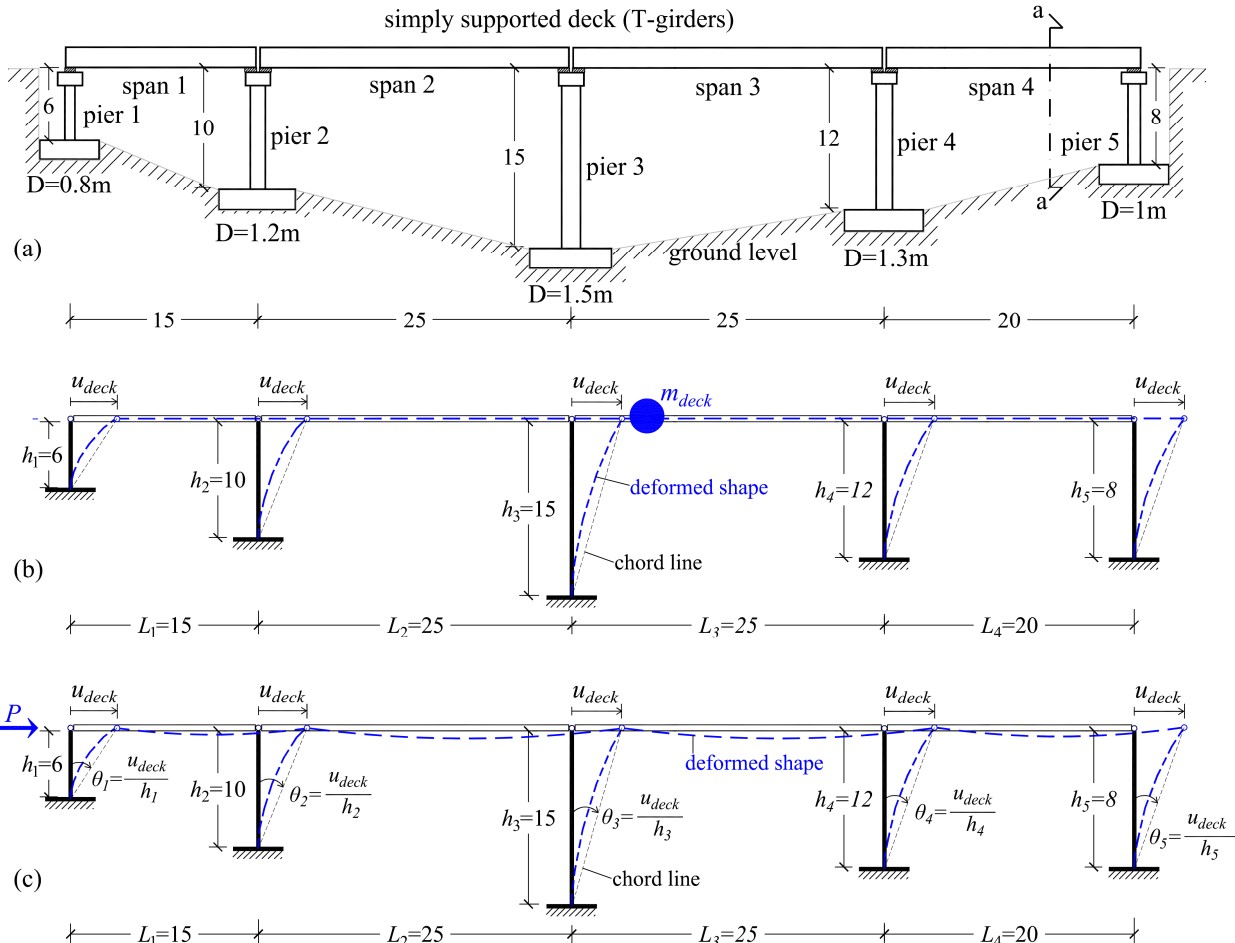

**Figure 7.** (**a**) Straight RC bridge with four spans and with (2 column) piers of various heights; (**b**) dynamic model of RC bridge (SDOF) for modal analysis; (**c**) static model of RC bridge for pushover analysis.

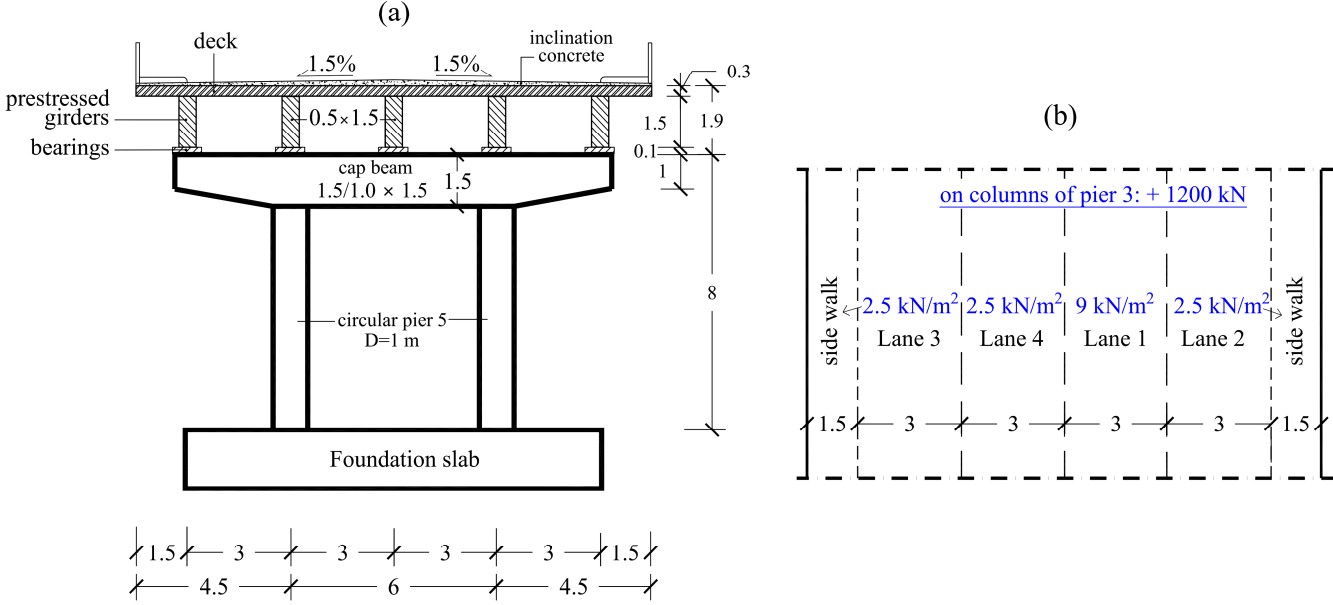

**Figure 8.** (**a**) Section aa at pier 5 of the RC bridge; (**b**) lane set-up and live loads on lanes.

The bridge accommodates four traffic lanes each with a width of 3 m and two sidewalks of 1.5 m (Figure 8b). The lane numbering and the live traffic loads on the lanes are detailed in Figure 8, following the LM1 load model of EN 1991-2 [35]. This load model encompasses uniformly distributed loads and includes a concentrated heavy vehicle load positioned to act at the most slender columns, specifically at the top of the two columns of pier 3. The total vertical loads $p = g + \psi_2 q$ in the seismic combination, where $g$ is the dead load, $\psi_E q$ is the quasi-live load, with $\psi_2 = 0.2$ for bridges [36], act at the top of the columns of the five piers and are outlined in Table 1. These loads contribute to a total bridge mass of approximately 2875 tons, assumed to be concentrated at the center of gravity of the deck system. The bridge mass at the deck and the degree of freedom along the longitudinal direction of the RC bridge for modal analysis are depicted in Figure 7b. Additionally, Figure 7c illustrates the static simulation of the RC bridge with the lateral force at the deck level for the pushover analysis.

**Table 1.** Longitudinal (L) and shear (h) reinforcement of circular columns.

| Pier Column | D (m) | Number of Bars | $d_{bL}$ (mm) | Geom. Ratio (%) | $d_{bh}$ (mm) | $s_h$ (cm) |
|---|---|---|---|---|---|---|
| 1 | 0.8 | 24 | 28 | 2.94 | 16 | 5 |
| 2 | 1.2 | 28 | 25 | 1.22 | 16 | 6 |
| 3 | 1.5 | 30 | 26 | 0.90 | 16 | 6 |
| 4 | 1.3 | 30 | 26 | 1.20 | 16 | 6 |
| 5 | 1 | 25 | 28 | 1.96 | 16 | 6 |

The construction materials of the RC bridge include concrete grade C35/40 with a mean compressive strength $f_{cm}$ of 43 MPa and steel grade B500c with a mean tensile strength $f_{ym}$ of 550 MPa. The modulus of elasticity for concrete is $E_c = 34$ GPa, while for steel, it is $E_s = 200$ GPa.

All circular column sections are symmetrically reinforced, with geometrical ratios ranging from 1% to 3% for the longitudinal reinforcement. The columns of the shortest pier 1 are the most reinforced. Confinement reinforcement in all columns consists of closed circular hoops with a diameter of 16 mm, evenly spaced at critical end sections with an axial spacing of 5 or 6 cm. The concrete cover is 5 cm. The steel reinforcement details of a typical circular column section are depicted in Figure 9. The longitudinal and shear reinforcements of the circular columns are outlined in Table 1. The reinforcement details of

the beam caps and the deck are omitted here as they are not relevant. The seismic behavior of the RC bridge relies exclusively on the behavior of the piers acting as cantilevers with a common top displacement, ensured by the rigid deck system. It is noteworthy that the RC bridge has been designed in accordance with EN 1998-1 [32] and EN 1998-2 [36] for the high ductility class (DCH). Consequently, it is expected to exhibit highly ductile behavior in the nonlinear domain, developing plastic hinges at the base of the columns.

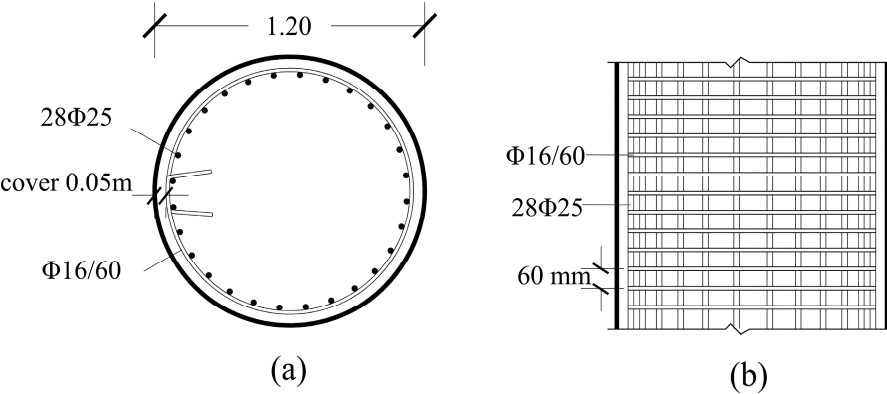

**Figure 9.** Pier 2: (**a**) section; (**b**) elevation.

Next, a series of pushover and instantaneous modal analyses is conducted to implement the "*M* and *P*" technique for structural damage identification in the existing RC bridge. The FEM analysis software SAP2000 [37] is utilized for this purpose. Plastic hinges of the fiber type are incorporated at the base of the piers in the nonlinear model of the bridge to simulate potential locations of inelastic behavior. The constitutive relationships used to characterize the behavior of construction materials in both the linear and nonlinear domains are as follows: (a) the model proposed by Mander, Priestley, and Park (1988) [38] for uniaxial unconfined and confined concrete (Figure 10); and (b) the model proposed by Park and Paulay (1975) [39] for steel reinforcement, which is parabolic in the strain hardening region (Figure 11). All necessary data for the nonlinear simulation of the bridge are derived from the results of a section analysis. The data for the base section of the piers are presented in Table 2, where the effective stiffness ratio $I_{eff}/I_g$ is calculated using Equation (7) in accordance with EN 1998-3 [33]. The plastic hinge length of the fiber hinges is calculated using Equation (8) and is shown in the last column of Table 2.

Next, as outlined in phase (c) of the "*M* and *P*" technique, Equations (9)–(11) from Figure 5 are employed to establish the stiffness scenario ($I_{eff}/I_g$) for the piers. This information is then incorporated into a series of pushover and instantaneous modal analyses of the bridge along the longitudinal direction, with a progressively increasing (target) deck displacement $u_{deck,i}$. The stiffness scenario is outlined in Table 3 as a function of the profile angle $\theta_{pr}$ at the base of the piers (Figure 7c). The discrete values of the effective moment of inertia $I_{eff}$ assigned to the two RC columns of each pier in the nonlinear model of the bridge depend on both the seismic (target) deck displacement of the pushover analysis and the pier height, i.e., on the target performance level of each pier (Figure 4). This is because the developed chord rotation at the base section of each pier $\theta_{c,i} = \theta_{pr,i} = u_{deck,i}/h_c$ at a target deck displacement $u_i$ depends on both these parameters and is different for each pier (Figure 7c). For example, if the target deck displacement $u_{deck,i}$ is equal to 0.3 m, then the developed chord rotations at the base sections of the two columns of each pier in rad units are: $\theta_1 = 0.3/6 = 0.05$, $\theta_2 = 0.3/10 = 0.03$, $\theta_3 = 0.3/15 = 0.02$, $\theta_4 = 0.3/12 = 0.025$, and $\theta_5 = 0.3/8 = 0.038$. Hence, different $I_{eff}/I_g$ values for the two columns of each pier for $u_{deck,i} = 0.3$ m are derived from Table 3, which are, respectively, equal to 0.32, 0.37, 0.39, 0.38, and 0.35.

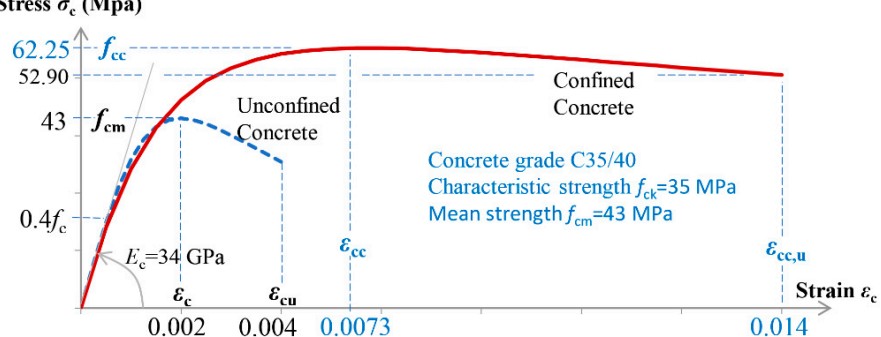

**Figure 10.** Stress–strain diagrams for both unconfined and confined concrete.

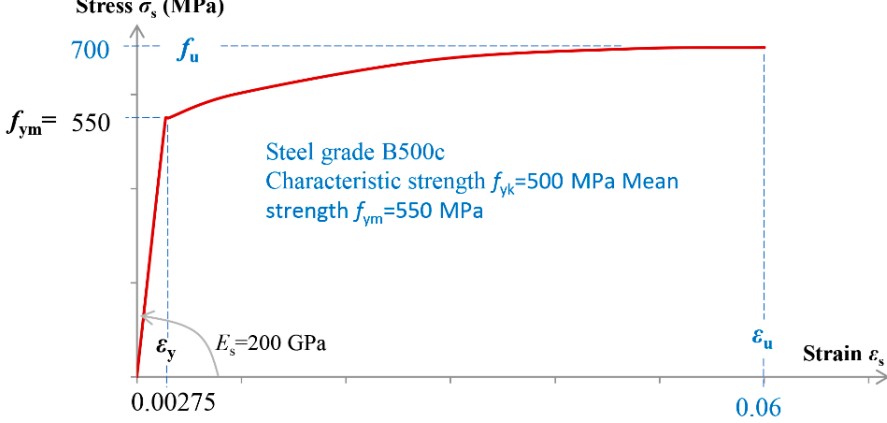

**Figure 11.** Stress–strain diagram for steel reinforcement bars.

**Table 2.** Base section analysis of RC pier columns to calculate the ratio ($I_{eff}/I_g$) from Equation (7), $E_c I_{eff} = M_p \cdot L_v / (3 \cdot \theta_y)$.

| Pier/ Column | Axial Force $N$ (kN) | $\nu = N/(A \cdot f_c)$ | $\varphi_y$ (rad/m) | $M_p$ (kNm) | $L_v$ (m) | $\theta_y$ (rad) | $E_c \cdot I_{eff}$ (kNm²) | $E_c \cdot I_g$ (kNm²) | $I_{eff}/I_g$ | $L_{pl}$ (m) |
|---|---|---|---|---|---|---|---|---|---|---|
| 1 | 1350 | 0.062 | 0.0098 | 2909 | 6 | 0.0264 | 220,531 | 683,611 | 0.32 | 0.62 |
| 2 | 3205 | 0.066 | 0.0053 | 5231 | 10 | 0.0226 | 773,128 | 3,460,778 | 0.22 | 0.80 |
| 3 | 4240 | 0.056 | 0.0040 | 8150 | 15 | 0.0243 | 1,679,135 | 8,449,166 | 0.20 | 1.04 |
| 4 | 3545 | 0.062 | 0.0049 | 6534 | 12 | 0.0243 | 1,074,144 | 4,766,748 | 0.23 | 0.90 |
| 5 | 1760 | 0.052 | 0.0069 | 4017 | 8 | 0.0241 | 444,408 | 1,668,971 | 0.27 | 0.72 |

**Table 3.** Effective stiffness scenario ($I_{eff}/I_g$) for the ductile RC pier columns as a function of base chord rotation $\theta_{pr}$ (in radians).

| $\theta_{pr}$ | $I_{eff}/I_g$ | $\theta_{pr}$ | $I_{eff}/I_g$ | $\theta_{pr}$ | $I_{eff}/I_g$ | $\theta_{pr}$ | $I_{eff}/I_g$ | $\theta_{pr}$ | $I_{eff}/I_g$ | $\theta_{pr}$ | $I_{eff}/I_g$ |
|---|---|---|---|---|---|---|---|---|---|---|---|
| 0 | 1.00 | 0.010 | 0.49 | 0.020 | 0.39 | 0.045 | 0.33 | 0.070 | 0.27 |
| 0.001 | 0.95 | 0.011 | 0.48 | 0.023 | 0.38 | 0.048 | 0.32 | 0.072 | 0.27 |
| 0.002 | 0.89 | 0.012 | 0.46 | 0.025 | 0.38 | 0.050 | 0.32 | 0.075 | 0.26 |
| 0.003 | 0.84 | 0.013 | 0.45 | 0.027 | 0.37 | 0.053 | 0.31 | 0.077 | 0.25 |
| 0.004 | 0.79 | 0.014 | 0.43 | 0.030 | 0.37 | 0.055 | 0.31 | 0.080 | 0.25 |
| 0.005 | 0.74 | 0.015 | 0.42 | 0.032 | 0.36 | 0.057 | 0.30 | 0.083 | 0.24 |
| 0.006 | 0.68 | 0.016 | 0.40 | 0.035 | 0.35 | 0.060 | 0.29 | 0.085 | 0.24 |
| 0.007 | 0.63 | 0.017 | 0.40 | 0.038 | 0.35 | 0.062 | 0.29 | 0.087 | 0.23 |
| 0.008 | 0.58 | 0.018 | 0.40 | 0.040 | 0.34 | 0.065 | 0.28 | 0.090 | 0.22 |
| 0.009 | 0.52 | 0.019 | 0.39 | 0.043 | 0.34 | 0.068 | 0.28 | 0.0903 | 0.22 |

Next, a pushover analysis is conducted to target the NC state, aiming to derive the capacity curve of the RC bridge along its longitudinal direction. The target deck

displacement causing the bridge NC state is set to 0.54 m. In this analysis, the effective moment of inertia ratio $I_{eff}/I_g$ of the columns of the various piers is determined from Table 3 (or from Figure 5) based on the corresponding values of $\theta_{pr}$ (in radians) that develop at their base sections at $u_{deck} = 0.54$ m. The resulting capacity curve of the RC bridge using the proposed stiffness scenario for the NC state is illustrated in Figure 12, along with the elastic-perfectly plastic idealization line, from which the yield displacement $u_y = 0.17$ m of the bridge results.

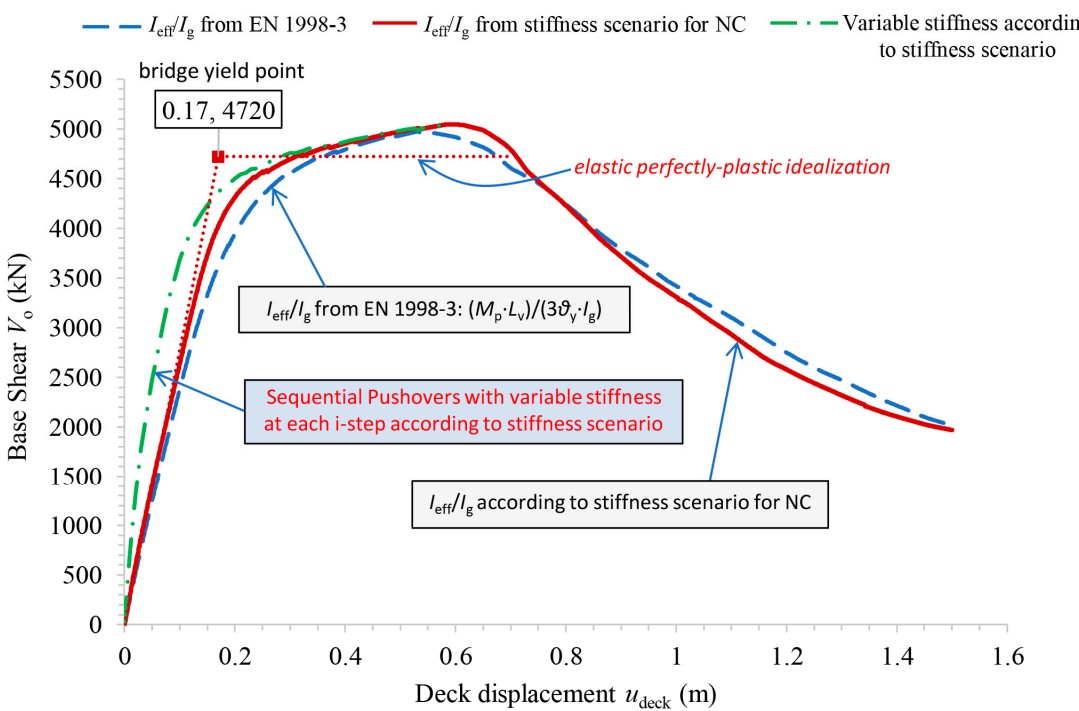

**Figure 12.** Capacity curve of the RC bridge along the longitudinal direction according to various stiffness scenarios. Ref. [33].

Additionally, Figure 12 includes the capacity curve of the bridge obtained from a pushover analysis using the stiffness scenario proposed in EN 1998-3 [33] (Equation (7)). Furthermore, Figure 12 displays the capacity curve of the bridge derived from sequential pushover analyses, targeting each time an increasing deck displacement from the health state to NC state, using the stiffness scenario proposed in Figure 5 for the discrete values of the target deck displacement. As observed in Figure 12, the influence of the effective stiffness scenario on the results of pushover analysis is more pronounced for higher performance levels due to the delayed onset of the yield state of the bridge. In Figure 13, the capacity curve of the bridge is presented again for the case of the proposed stiffness scenario at the NC state, along with the capacity curves of the various piers (each having two columns). The contribution of the various piers to the capacity of the bridge, in terms of base shear and deck displacement, is evident in the figure. The yield points of the piers, derived from the bilinear idealization of their curves, are also displayed in Figure 13, illustrating the yield sequence. The NC state of the bridge is induced by the failure of the shortest columns of pier 1, which develop a chord rotation $\theta_{pr}$ at their base sections equal to $\theta_1 = 0.54/6 = 0.09$ rad. At the failure of the columns of pier 1, the columns of the other piers develop the following chord rotations at their base sections: $\theta_2 = 0.54/10 = 0.054$, $\theta_3 = 0.54/15 = 0.036$, $\theta_4 = 0.54/12 = 0.045$, and $\theta_5 = 0.54/8 = 0.068$ in rad units. It is noted that for lower values of the target deck displacement driving the RC bridge to higher performance levels (for example, DL or SD), the corresponding values of $I_{eff}/I_g$ from Table 3 should be assigned to the piers. Therefore, the capacity curve resulting from the

pushover analysis of the bridge targeting a higher performance level will not exhibit exactly the same characteristics as that for the NC state (see Figure 14).

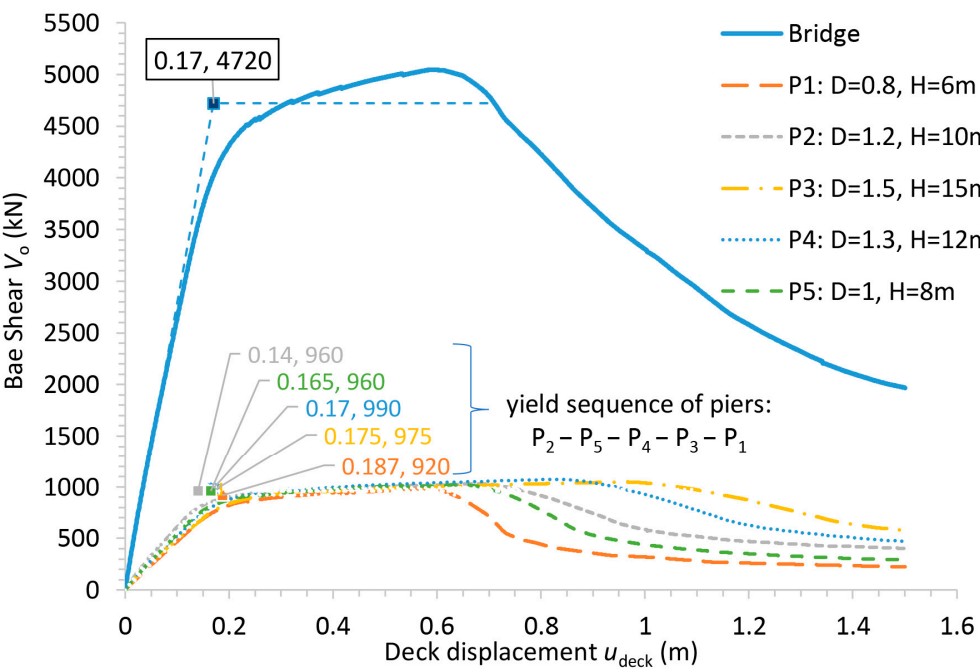

**Figure 13.** Capacity curve of the RC bridge and of the various (2 columns) piers along the longitudinal direction according to the stiffness scenario for the NC state.

Next, a series of pushover analyses is performed on the existing RC bridge along its longitudinal direction. Each analysis targets an increasing deck displacement corresponding to the specific values of the profile angle $\theta_{pr}$ of Table 3, ranging from 0 to 0.09 rad. The initial pushover analysis in the series targets the health (uncracked) state of the bridge. Therefore, it is conducted with a lateral (target) deck displacement set to zero and considers only the effect of the vertical loads in the seismic combination, $g + \psi_E q$. For this analysis, all piers in the nonlinear model of the RC bridge are provided with the value $I_{eff} = I_g$. In all subsequent pushover analyses in the series, the various piers of the bridge are provided with the effective moments of inertia $I_{eff,i}$ from Table 3 (or from Figure 5), corresponding to the developed values of the profile angle $\theta_{pr,i}$ at the base section of the pier columns at the target deck displacement $u_i$.

Subsequently, a series of instantaneous modal analyses along the longitudinal direction of the RC bridge is performed following the final step of each one of the separate pushover analyses in the series. These modal analyses have as initial condition the damage state of the bridge at the final step of the separate pushover analyses with target deck displacement $u_i$. In other words, the stiffness of the damaged bridge at the final step of each pushover analysis is used. From this series of modal analyses on the RC bridge, the instantaneous cyclic eigenfrequency $f_i$ (in Hz) of the SDOF system along the longitudinal direction is recorded at each *i*-step. This information is then utilized to generate the diagram of the instantaneous cyclic eigenfrequency (in Hz) of the RC bridge across the linear and nonlinear domains as a function of the deck displacement, $u_{deck,i}$. Figure 14 depicts this diagram along with an approximate line that encompasses all performance levels.

The cyclic eigenfrequency $f_i$ (in Hz) resulting from the instantaneous modal analyses at the final step of each separate pushover analysis are presented in Table 4 for indicative values of the target displacement $u_{deck,i}$. These data correspond to the information depicted in Figure 14.

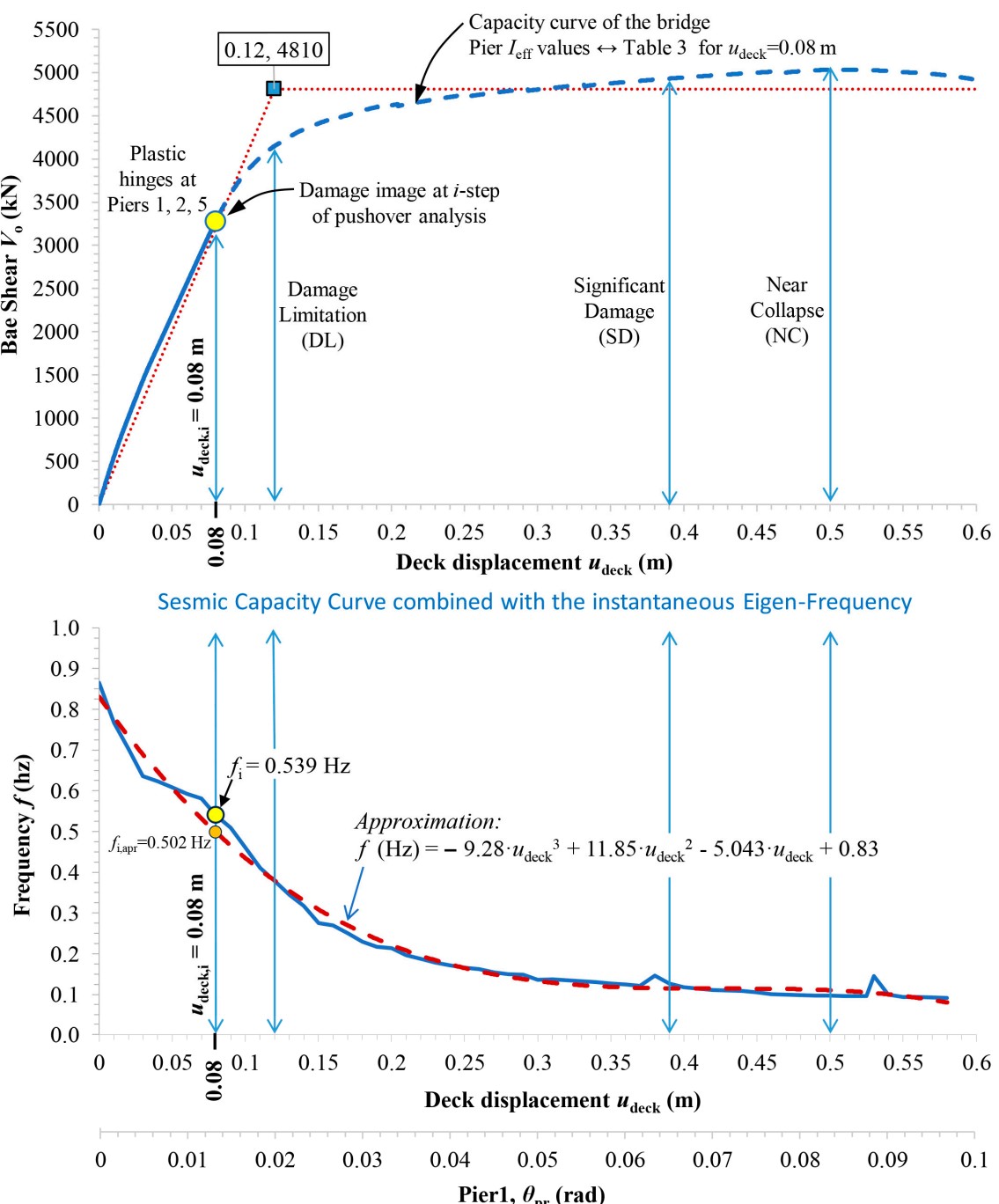

**Figure 14.** Instantaneous eigenfrequency diagram combined with the seismic capacity curve of the RC bridge along the longitudinal direction (key diagram).

As evident from Table 4, the instantaneous period $T_i$ (in seconds) of the existing (damaged) RC bridge—corresponding to the inverse of the cyclic eigenfrequency $f_i$—is elongated with increasing deck displacement. This indicates that the RC bridge gradually becomes more flexible due to the advancing damage. The elongation of the period between the health state and the DL state is observed to be more than two times. Figure 14 presents the diagram of the instantaneous cyclic eigenfrequency $f_i$ (in Hz) in conjunction with the capacity curve of the RC bridge along the longitudinal direction. These two diagrams are interconnected and form the *key-diagram* of the proposed "*M* and *P*" technique for bridges. This key-diagram serves as a crucial tool in the "*M* and *P*" technique, facilitating the identification and assessment of damage in the bridge structure. In this key-diagram:

(a)    The monitoring fundamental frequency of the existing bridge is inserted (see Figure 2), and the deck seismic displacement $u_{deck,i}$ (or the corresponding $\theta_{pr}$ value of the piers) of the existing RC bridge is determined.

(b)    At this seismic (target) deck displacement $u_{deck,i}$, we ascend to the capacity curve of the RC bridge, allowing us to visually observe the damage state.

**Table 4.** Instantaneous cyclic eigenfrequency $f_i$ (Hz) of the RC bridge (Figure 14), from the health state to the Near Collapse (NC) state.

| $u_{deck}$ (m) | $\theta_{pr}$ of the Column of Piers in Rad | | | | | $I_{eff}/I_g$ | | | | | State of Damage | $f$ (hz) | $T$ (s) |
|---|---|---|---|---|---|---|---|---|---|---|---|---|---|
| | $P_1$ | $P_2$ | $P_3$ | $P_4$ | $P_5$ | $P_1$ | $P_2$ | $P_3$ | $P_4$ | $P_5$ | | | |
| 0 | 0.000 | 0.000 | 0.000 | 0.000 | 0.000 | 1 | 1 | 1 | 1 | 1 | Health | 0.865 | 1.16 |
| 0.04 | 0.007 | 0.004 | 0.003 | 0.003 | 0.005 | 0.65 | 0.79 | 0.86 | 0.82 | 0.74 | | 0.623 | 1.61 |
| 0.07 | 0.012 | 0.007 | 0.005 | 0.006 | 0.009 | 0.47 | 0.63 | 0.75 | 0.69 | 0.54 | 1st Hinge | 0.581 | 1.72 |
| 0.08 | 0.013 | 0.008 | 0.005 | 0.007 | 0.010 | 0.44 | 0.58 | 0.72 | 0.65 | 0.49 | | 0.539 | 1.86 |
| 0.09 | 0.015 | 0.009 | 0.006 | 0.008 | 0.011 | 0.42 | 0.52 | 0.68 | 0.60 | 0.47 | | 0.509 | 1.97 |
| 0.1 | 0.017 | 0.010 | 0.007 | 0.008 | 0.013 | 0.40 | 0.49 | 0.65 | 0.56 | 0.45 | | 0.461 | 2.17 |
| 0.12 | 0.020 | 0.012 | 0.008 | 0.010 | 0.015 | 0.39 | 0.46 | 0.58 | 0.49 | 0.42 | DL | 0.379 | 2.64 |
| 0.14 | 0.023 | 0.014 | 0.009 | 0.012 | 0.018 | 0.38 | 0.43 | 0.51 | 0.47 | 0.40 | | 0.318 | 3.15 |
| 0.18 | 0.030 | 0.018 | 0.012 | 0.015 | 0.023 | 0.37 | 0.40 | 0.46 | 0.42 | 0.38 | | 0.229 | 4.36 |
| 0.2 | 0.033 | 0.020 | 0.013 | 0.017 | 0.025 | 0.36 | 0.39 | 0.44 | 0.40 | 0.38 | | 0.213 | 4.69 |
| 0.22 | 0.037 | 0.022 | 0.015 | 0.018 | 0.028 | 0.35 | 0.39 | 0.42 | 0.39 | 0.37 | | 0.188 | 5.33 |
| 0.25 | 0.042 | 0.025 | 0.017 | 0.021 | 0.031 | 0.34 | 0.38 | 0.40 | 0.39 | 0.36 | | 0.166 | 6.03 |
| 0.27 | 0.045 | 0.027 | 0.018 | 0.023 | 0.034 | 0.33 | 0.37 | 0.40 | 0.38 | 0.36 | | 0.154 | 6.48 |
| 0.3 | 0.050 | 0.030 | 0.020 | 0.025 | 0.038 | 0.32 | 0.37 | 0.39 | 0.38 | 0.35 | | 0.137 | 7.33 |
| 0.33 | 0.055 | 0.033 | 0.022 | 0.028 | 0.041 | 0.31 | 0.36 | 0.39 | 0.37 | 0.34 | | 0.133 | 7.54 |
| 0.35 | 0.058 | 0.035 | 0.023 | 0.029 | 0.044 | 0.30 | 0.35 | 0.38 | 0.37 | 0.33 | | 0.127 | 7.87 |
| 0.37 | 0.062 | 0.037 | 0.025 | 0.031 | 0.046 | 0.29 | 0.35 | 0.38 | 0.36 | 0.33 | | 0.121 | 8.28 |
| 0.4 | 0.067 | 0.040 | 0.027 | 0.033 | 0.050 | 0.28 | 0.34 | 0.37 | 0.36 | 0.32 | | 0.118 | 8.49 |
| 0.42 | 0.070 | 0.042 | 0.028 | 0.035 | 0.053 | 0.27 | 0.34 | 0.37 | 0.35 | 0.31 | SD | 0.111 | 9.03 |
| 0.45 | 0.075 | 0.045 | 0.030 | 0.038 | 0.056 | 0.26 | 0.33 | 0.37 | 0.35 | 0.30 | | 0.105 | 9.49 |
| 0.47 | 0.078 | 0.047 | 0.031 | 0.039 | 0.059 | 0.25 | 0.33 | 0.36 | 0.34 | 0.30 | | 0.100 | 10.00 |
| 0.5 | 0.083 | 0.050 | 0.033 | 0.042 | 0.063 | 0.24 | 0.32 | 0.36 | 0.34 | 0.29 | NC | 0.097 | 10.32 |

The eigenvalue curve of Figure 14 was generated using a nonlinear model of the RC bridge in which the effective moment of inertia values $I_{eff}$ for the piers were obtained from Table 1 (or from Figure 5) for a deck target displacement $u_{deck,i} = 0.08$ m. At this target displacement, the resulting chord rotations $\theta_{pr}$ at the base section of the columns for piers 1 to 5 are as follows: $\theta_1 = 0.08/6 = 0.0133$, $\theta_2 = 0.08/10 = 0.008$, $\theta_3 = 0.08/15 = 0.00533$, $\theta_4 = 0.08/12 = 0.0067$, and $\theta_5 = 0.08/8 = 0.01$ rad. Therefore, the corresponding $I_{eff}/I_g$ values for piers 1 to 5 are 0.44, 0.58, 0.72, 0.65, and 0.49, respectively (Table 4). These values contribute to the accurate representation of the bridge's dynamic behavior as depicted in the eigenvalue curve.

According to phase (a) of the "*M* and *P*" technique for existing bridges, an identification monitoring system is installed in the RC bridge, and response accelerations along the longitudinal direction are recorded when the bridge is in a quasi-calm state. The analysis of the records reveals an identified eigenfrequency $f_i = 0.539$ Hz for the *i*-step of the pushover analysis. Subsequently, in accordance with phase (d) of the "*M* and *P*" technique, the identified eigenfrequency $f_i$ is inserted into the eigenfrequency diagram of Figure 14, enabling the determination of the corresponding displacement $u_i = 0.08$ m of the bridge deck. This displacement corresponds to specific $\theta_{pr}$ values for each pier, as mentioned above. Additionally, at the final step of the pushover analysis targeting the deck displacement $u_i = 0.08$ m, a visual representation of the damage state in the RC bridge is acquired (damage image). As illustrated in Figure 15, the columns of piers 1, 2, and 5 have just yielded at approximately the same displacement. Figure 16 displays the capacity

curves of the bridge piers resulting from a pushover analysis targeting a deck displacement $u_{deck} = 0.08$ m, in terms of base shear and $\theta_{pr}$.

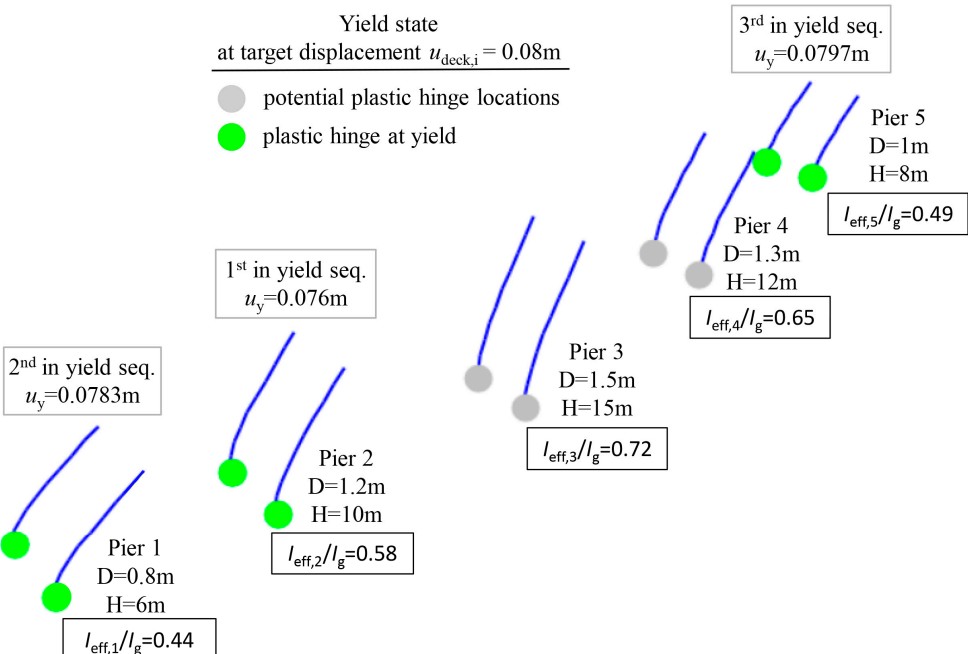

**Figure 15.** Yield state and sequence of yield of the bridge piers at deck target displacement $u_i = 0.08$ m.

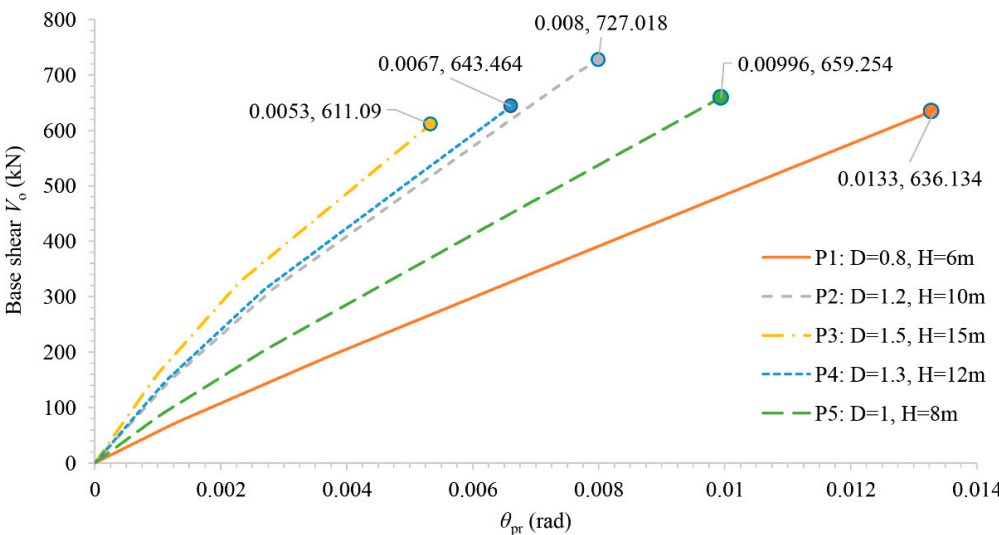

**Figure 16.** Capacity curve of the RC bridge piers for a deck target displacement $u_{deck} = 0.08$ m, in terms of base shear and $\theta_{pr}$.

The eigenfrequency of the bridge $f_i = 0.539$ Hz, corresponding to a deck seismic displacement $u_{deck,i} = 0.08$ m, is approximately 63% of the eigenfrequency $f_{Health} = 0.865$ Hz at the health (uncracked) state of the bridge. This presents an elongation in period values from $T_{Health} = 1.156$ s to $T_i = 1.856$ s at this damage state.

It is emphasized that in a seismic event, the actual seismic load on the bridge along the longitudinal direction is different and varies at each time step. As a result, the damage distribution on the bridge and the cyclic degradation of stiffness, closely linked to the duration and number of cycles of the ground excitation, may slightly differ from that obtained through pushover analysis. However, the critical parameter in the current proposed methodology for bridges is the eigenfrequency $f_i$, identified by the monitoring procedure with a local network of uniaxial accelerometers under calm conditions. With the knowl-

edge of the eigenfrequency $f_i$ of the bridge, the seismic lateral deck displacement $u_{deck,i}$ along the longitudinal direction of the bridge can be estimated, as illustrated in Figure 14. Subsequently, the capacity curve is utilized to identify the damage state of the bridge piers.

In the final phase (g) of the "$M$ and $P$" technique, the instantaneous stiffness $k_i$ along the longitudinal direction of the RC bridge is computed at the examined inelastic $i$-step where the deck displacement is equal to 0.08 m. To achieve this, a linear analysis is conducted with a prescribed lateral force $F_i$ applied at the deck level, following the final step of the pushover analysis where the deck displacement $u_{deck,i} = 0.08$ m is reached. The corresponding static displacement $u_{st,i}$ is then calculated. Next, the stiffness $k_i$ of the damaged bridge is calculated using the ratio $F_i/u_{st,i}$. Therefore, the damage stiffness $\Delta k_i$ of the bridge at the same inelastic $i$-step is derived from the general relationship $\Delta k_i = k_o - k_i$, where $k_o$ represents the known initial stiffness of the undamaged bridge, which is calculated at the health state using the same procedure. Additionally, the damage stiffness $\Delta k_{c,i}$ of the various RC piers of the bridge can be obtained from the preceding linear analysis by recording their base shear. Consequently, the location and magnitude of the damage at the examined seismic deck displacement $u_{deck,i} = 0.08$ m can be identified.

The damage stiffness of the RC bridge and that of the bridge piers at the seismic deck displacement $u_{deck,i} = 0.08$ m is provided in Table 5. By knowing the damage stiffness $\Delta k_i$ and $\Delta k_{c,i}$ of the RC bridge and the various RC piers along the longitudinal direction of the bridge, the final percentage deviation terms of $\Delta k_i$ and $\Delta k_{c,i}$ can be calculated with respect to the initial stiffness $k_o$ and $k_{c,o}$, respectively. These deviation terms are presented in the last column of Table 5. A visual representation of this table is depicted in Figure 17.

**Table 5.** Percentage deviation of the damage stiffness $\Delta k_i$ for the RC bridge and the bridge piers at the inelastic $i$-step corresponding to the seismic deck displacement $u_{deck,i} = 0.08$ m.

| Pier with 2 Columns | Health State $k_o$ (kN/m) | Damage State at $u_{deck,i} = 0.08$ m | | |
| --- | --- | --- | --- | --- |
| | | $k_i$ (kN/m) | $\Delta k_i = k_o - k_i$ (kN/m) | $\Delta k_i/k_o$ (%) |
| C1a | 8932 | 3330 | 5602 | 63 |
| C1b | 8932 | 3330 | 5602 | 63 |
| C2a | 9625 | 2819 | 6806 | 71 |
| C2b | 9625 | 2819 | 6806 | 71 |
| C3a | 7025 | 3107 | 3919 | 56 |
| C3b | 7025 | 3107 | 3919 | 56 |
| C4a | 7724 | 3498 | 4226 | 55 |
| C4b | 7724 | 3498 | 4226 | 55 |
| C5a | 9141 | 3711 | 5430 | 59 |
| C5b | 9141 | 3711 | 5430 | 59 |
| Bridge | 84,895 | 32,930 | 51,965 | 61 |

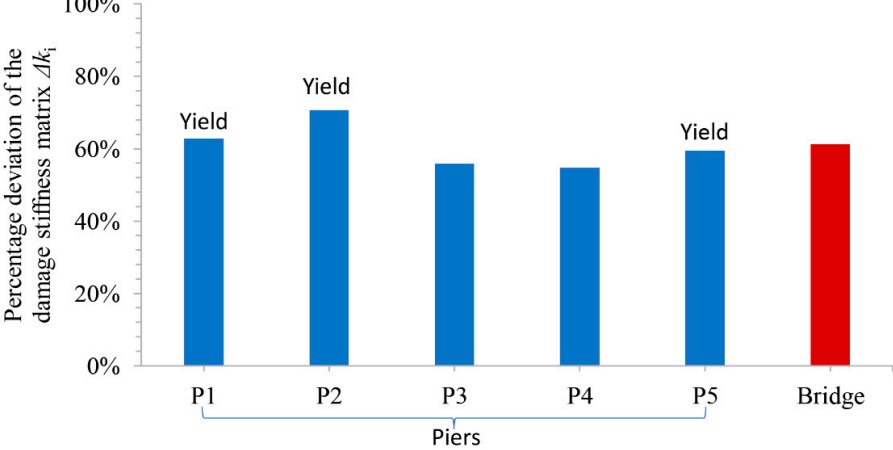

**Figure 17.** Percentage deviation of the damage stiffness $\Delta k_i$ for the RC bridge and the bridge piers at the inelastic $i$-step corresponding to the seismic deck displacement $u_{deck,i} = 0.08$ m.

## 4. Discussion

As indicated by the results presented in Table 4 for the eigenfrequency, in Figure 14 for the deck (target) displacement, in Figures 15 and 17 for the plastic mechanism, as well as in Table 5 and Figure 17 for the damage stiffness of the existing RC bridge, the location and severity of the damage in the RC bridge can now be confirmed in relation to the health state. This confirmation is feasible due to the interconnected nature of all the aforementioned parameters. This achievement aligns with the objectives of the proposed "*M* and *P*" hybrid technique for bridges.

Indeed, by computing the damage stiffness $\Delta k_i$ of the RC bridge along the longitudinal direction, and, additionally, the damage stiffness $\Delta k_{c,i}$ of the various bridge piers, the final percentage deviation terms of $\Delta k_i$ and $\Delta k_{c,i}$ can be determined with respect to the initial stiffness $k_o$ and $k_{c,o}$ at the health state (Table 5 and Figure 17). These deviations in damage stiffness express the extent of damage in the RC bridge as a whole and at the base section of the bridge piers, at the final step of pushover analysis where the seismic deck displacement $u_{deck,i} = 0.08$ m occurs. This aligns seamlessly with the damage image in Figure 15. This article addresses the identification of damage along the longitudinal direction of straight, ductile, RC bridges with rigid decks. In this type of existing bridges, damage always occurs at the base sections of the columns of the various bridge piers. It is important to note that the stiffness terms of the various piers correspond to the lateral dynamic degree of freedom of the SDOF RC bridge along the longitudinal direction (Figure 7). Therefore, the deviation of these stiffness terms $\Delta k_{c,i}$ relative to the health state corresponds to the overall damage of the piers at their critical base region, which occurs at a discrete seismic target deck displacement. From Table 5 and Figure 17, it is observed that the circular columns of the piers 1, 2, and 5 exhibit higher deviation terms in damage stiffness $\Delta k_{c,i}$ compared to piers 3 and 4. Hence, the stiffness terms $\Delta k_{c,i}$ of the piers, as well as their magnitude, which constitute the total stiffness degradation $\Delta k_i$ of the RC bridge along the longitudinal direction, are fully consistent with the damage image in Figure 15. This image illustrates that the base sections of these piers present the greatest damage.

Hence, it is demonstrated that, given a specific damage image in an existing, ductile, RC bridge, the stiffness of the bridge undergoes changes relative to the health state. This alteration in stiffness leads to a shift in the eigenfrequency of the bridge, which is experimentally identified through the monitoring procedure. Subsequently, utilizing the key diagram of the proposed "*M* and *P*" technique for bridges, the seismic deck displacement of the bridge along the longitudinal direction is determined. This displacement, on the one hand, aligns with the observed damage image and, on the other hand, it ensures the eigenfrequency matches the field-measured value. All the parameters utilized in the hybrid "*M* and *P*" technique for bridges are interconnected, establishing a self-evident and accurate methodology.

## 5. Conclusions

A new proposed hybrid technique for the identification of damage in ductile, RC bridges is evaluated in the current paper by investigating a group of straight, multiple-span bridges with rigid decks and piers of various heights, from which a specific numerical example was presented here. This is a four-span RC bridge with five piers, each consisting of two circular columns behaving as cantilevers.

The newly introduced hybrid technique for bridges, referred to as the "*M* and *P* technique" (where *M* stands for "Monitoring" and *P* for "Pushover"), integrates the pushover capacity curve of the bridge along its longitudinal direction with the diagram illustrating the instantaneous eigenfrequency of the bridge in relation to the inelastic seismic (target) deck displacement. This key diagram was generated through a series of pushover and instantaneous modal analyses, where the target deck displacement $u_{deck,i}$ gradually increased, corresponding to specific values of the chord rotations $\theta_{pr,i}$ of the bridge piers. In each analysis, the circular columns of the bridge piers in the nonlinear model are assigned suitable values of the effective bending stiffness $E_c I_{eff,i}$ corresponding to the target

deck displacement. This is accomplished following a proposed stiffness scenario based on the principles outlined in EN1998-3. By incorporating the eigenfrequency of the existing (damaged) RC bridge in this diagram, initially identified through a network of accelerograms in the monitoring phase, the target deck displacement $u_{deck,i}$ of the bridge arises. Consequently, the damage image of the bridge at the final step of the pushover analysis targeting this specific deck displacement is revealed. Moreover, the instantaneous stiffness and the damage stiffness of the entire RC bridge, as well as the corresponding values of the individual bridge piers are computed at this final step of pushover analysis where the target displacement $u_{deck,i}$ is observed. The damage stiffness of the various bridge piers is fully compatible with the extent of damage at the base section of the corresponding piers.

The interconnected nature of the parameters involved in the proposed hybrid "*M* and *P*" technique for bridges, including deck displacement, bridge stiffness, and eigenvalue, enables a highly accurate estimation of the damage pattern in an existing RC bridge. This is achieved because all the interconnected parameters are derived from the eigenfrequency measured in the field by the monitoring procedure. The technique consistently predicts the location and severity of damage among the various bridge piers, providing a comprehensive understanding of the bridge's health.

While the "*M* and *P*" technique is currently well-suited for damage identification along the longitudinal direction of a straight RC bridge, ongoing research is exploring its applicability for detecting damage along the direction perpendicular to the bridge axis. Furthermore, investigations are underway to assess its effectiveness in other bridge geometric forms, including skewed or curved bridges. To broaden the scope of the current research methodology, further exploration into the method's applicability should consider investigating the vertical component of earthquakes and examining the complexities introduced by soil–structure interaction. Additionally, future investigations will focus on integrating the hybrid "*M* and *P*" technique into health monitoring procedures and exploring its potential use in neural networks.

**Author Contributions:** Conceptualization, A.B. and T.M.; methodology, A.B. and T.M.; software, A.B.; validation, A.B., T.M. and V.L.; formal analysis, A.B. and T.M.; investigation, A.B. and T.M.; resources, A.B., T.M. and V.L.; data curation, A.B., T.M. and V.L.; writing—original draft preparation, A.B.; writing—review and editing, A.B., T.M. and V.L.; visualization, A.B. and T.M.; supervision, A.B., T.M. and V.L.; project administration, A.B. and T.M.; funding acquisition, None. All authors have read and agreed to the published version of the manuscript.

**Funding:** This research received no external funding.

**Data Availability Statement:** Data presented in this study are available in the article.

**Conflicts of Interest:** The authors declare no conflicts of interest.

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
