# Peer review of "The “M and P” Technique for Damage Identification in Reinforced Concrete Bridges"

_infrastructures, doi:10.3390/infrastructures9020018_

Round 1
Reviewer 1 Report
Comments and Suggestions for Authors
This article covers an interesting topic in the research field of earthquake engineering.
My recommendation is that this paper is suitable for publication after minor revision.
Specific comments:
1.
Describe any existing limitations of the proposed procedure.
2.
Comment on the possible extensibility of the proposed procedure in future research, including the effect of vertical seismic components or the soil-structure interaction.
3.
The correct SI-metric symbol for "seconds" is "s" and not "sec".
4.
It is written in the introduction section that "… originally developed by Makarios [33, 34]". However, Makarios is not an author of the references [33, 34].
Author Response
Initially we want to thank all the honorable Reviewers for their contribution to our paper.
Detailed responses are presented below to all reviewer's comments.
Two versions of the revised paper have been prepared. The first version includes all the corrections made in the manuscript, with the originally highlighted erased text and the new corrected or inserted text presented in red font. The second version includes only the corrected text and the newly inserted text, both highlighted in red font.
It is noted that, due to the insertion of new text, particularly in the Introduction, Methodology, and References sections, the line numbering no longer aligns with the initially submitted manuscript.
Answers to Reviewers
Reviewer 1
This article covers an interesting topic in the research field of earthquake engineering.
My recommendation is that this paper is suitable for publication after minor revision.
Specific comments:
1.
Describe any existing limitations of the proposed procedure.
Answer:
The presented paper demonstrates the proposed M and P procedure for damage identification in reinforced concrete (RC) bridges. The focus is on identifying damage along the longitudinal direction of a multi-span, ductile RC bridge with a rigid deck, where beam girders are simply supported on pier caps, and the piers vary in height (or are of the same height).
This is clear in Introduction (L.88-92, 111-116), in Methodology (L.134-143, 144-145, 199-200), in Numerical example (L.420-437) and in Conclusions (L.734-738, 766-767). In this study, only straight bridges were investigated.
The primary objective of the current paper is to demonstrate the efficacy of the proposed M and P technique in damage detection, employing the key-diagram of Fig.14. This involves integrating a monitoring system with pushover analysis.
The term ‘’straight” was added in Abstract, to indicate that straight bridges were examined. It is also added in Introduction L.89, L.114, in Methodology L.136, in Figure 3 caption, in Numerical example L.421, in Figure 7 caption, in Discussion L.711, in Conclusions L.736, L.768.
2.
Comment on the possible extensibility of the proposed procedure in future research, including the effect of vertical seismic components or the soil-structure interaction.
Answer:
We agree with the reviewer. The future investigation will consider the impact of the vertical seismic component and address the complexities introduced by soil-structure interaction.
This is added at the end of Conclusions.
Also, investigation is underway to assess the effectiveness in other bridge geometric forms, such as skewed or curved bridges.
3.
The correct SI-metric symbol for "seconds" is "s" and not "sec".
Answer:
Corrected in Table 4 heading and in line 649.
4.
It is written in the introduction section that "… originally developed by Makarios [33, 34]". However, Makarios is not an author of the references [33, 34].
Answer:
The correct references are [25,26], which are renumbered now as [30, 31] due to the insertion of new references above. Corrected in text.
Reviewer 2 Report
Comments and Suggestions for Authors
The study concerns the proposal of a methodology for the use of sensor-based frequency detection for predicting the seismic demand (and damage) in terms of deck displacement. Several limitations and criticalities are associated to the proposed approach and manuscript.
1. The method is basically applicable for multi-span bridges analysed in the longitudinal direction. Therefore, the application field of the methodology is very limited, considering that the transverse direction can be the most critical in multi-span bridge classes. In addition, the authors should clarify if the method is applicable for skewed and curve bridges.
2. Only the damage state of piers is considered. The authors should discuss the potential of the method to predict the damage of bearing devices and abutment-backfill system which can be critical in non-seismically designed bridges (https://doi.org/10.1007/s10518-023-01783-y, https://doi.org/10.1007/s10518-008-9077-2).
3. L109 the authors mention “the damage image”. This term is not clear.
4. As explained at L133, the authors model the response of the bridge in longitudinal direction as a SDoF system. Other studies adopting this approach should be mentioned (e.g. https://doi.org/10.1007/s10518-023-01783-y).
5. L305 the authors mention results of an “extended parametric analysis” which is carried out to retrieve relations between the effective stiffness and the chord rotation. Further information are required for the reader to understand the applicability of the proposed relations.
6. L307 “cyclic section”. Please clarify.
7. The stiffness (and the frequency) of a seismically damaged structure depends not only on the seismic deck displacement, but also on the cyclic degradation which can be related to the duration and number of cycles of the experienced ground excitation. In other words, for the same seismic target displacement, the “damaged” stiffness (and the associated modal frequency) will depend on the number of cycles experienced. The higher the number of cycles in the non-linear range, the greater the stiffness degradation and the greater the period increase, even if the maximum deck displacement is constant. Therefore, considering only the deck displacement, the proposed method ca lead to erroneous estimation. The authors should comment on this point.
8. Figure 6: correct the text in the rhombus.
9. The authors state that appropriate values of the effective stiffness should be assigned to the numerical model. However, the use of a distributed plasticity model (several plastic hinges discretised along the pier height) can be useful to avoid the effective stiffness estimations. Comments on this point are required.
Author Response
Initially we want to thank all the honorable Reviewers for their contribution to our paper.
Detailed responses are presented below to all reviewers’ comments.
Two versions of the revised paper have been prepared. The first version includes all the corrections made in the manuscript, with the originally highlighted erased text and the new corrected or inserted text presented in red font. The second version includes only the corrected text and the newly inserted text, both highlighted in red font.
It is noted that, due to the insertion of new text, particularly in the Introduction, Methodology, and References sections, the line numbering no longer aligns with the initially submitted manuscript.
Answers to Reviewers
Reviewer 2
The study concerns the proposal of a methodology for the use of sensor-based frequency detection for predicting the seismic demand (and damage) in terms of deck displacement. Several limitations and criticalities are associated to the proposed approach and manuscript.
- The method is basically applicable for multi-span bridges analysed in the longitudinal direction. Therefore, the application field of the methodology is very limited, considering that the transverse direction can be the most critical in multi-span bridge classes. In addition, the authors should clarify if the method is applicable for skewed and curve bridges.
Answer:
Indeed, as mentioned clearly at various sections in the manuscript [in Introduction (L.88-92, 111-116), in Methodology (L.134-143, 144-145, 199-200), in Numerical example (L.420-437) and in Conclusions (L.734-738, 766-767)], the proposed M and P procedure is focused in this paper on identifying damage along the longitudinal direction of a multi span, ductile, RC bridge, considering rigid deck behavior with beam girders simply supported on pier caps (adjacent independent bearing-supported prestressed girders) and with piers of various heights.
The behavior of the bridge in the transverse direction to the bridge axis was not explored in this study. The primary objective of the current paper is to demonstrate the efficacy of the proposed M and P technique in damage detection, employing the key-diagram of Fig.14. This involves integrating a monitoring system with pushover analysis.
In our investigation, the analysis was limited to straight bridges. Skewed and curve bridges were not investigated but this issue is under investigation. This clarification is added into the manuscript. The term ‘’straight” was added in Abstract, to indicate that straight bridges were examined. It is also added in Introduction L.89, L.114, in Methodology L.136, in Figure 3 caption, in Numerical example L.421, in Figure 7 caption, in Discussion L.711, in Conclusions L.736, L.768.
In Conclusions (lines 766-775), we refer to the future extension of the proposed methodology concerning the investigation along the perpendicular direction to the bridge axis, as well as its application to other bridge geometric forms.
- Only the damage state of piers is considered. The authors should discuss the potential of the method to predict the damage of bearing devices and abutment-backfill system which can be critical in non-seismically designed bridges (https://doi.org/10.1007/s10518-023-01783-y, https://doi.org/10.1007/s10518-008-9077-2).
Answer:
The current investigation centers on damage identification along the longitudinal direction of straight RC bridges with a rigid deck, utilizing the hybrid M and P technique. This approach integrates a monitoring system with pushover analysis (key diagram). Our objective is to showcase the effectiveness of the proposed methodology in identifying damage. Only the damage state of the piers, which behave as cantilevers, was considered in our work. The deck system (with prestressed girders simply supported on pier caps) was considered rigid.
The behavior of bearing devices was not taken into account in our work, as the bearings were treated merely as simple support areas for the bridge girders. Additionally, the impact of the abutment-backfill system on the bridge behavior was not taken into consideration.
Certainly, all these parameters that influence bridge inelastic behavior, particularly in non-seismically designed bridges, could be subject to further investigation. In a future investigation, the behavior of bearing devices and the influence of an abutment-backfill system could be integrated into the nonlinear model of the bridge. Details regarding this integration, as well as the modeling of the bridge response in the longitudinal direction as a Single Degree of Freedom (SDoF) system (from your comment 4 below), are provided in the attached references.
These references will be added in Introduction in L.70.
[25] Nettis, A., Raffaele, D. & Uva, G. Seismic risk-informed prioritisation of multi-span RC girder bridges considering knowledge-based uncertainty. Bull Earthquake Eng (2023). https://doi.org/10.1007/s10518-023-01783-y
[26] Moschonas, I.F., Kappos, A.J., Panetsos, P. et al. Seismic fragility curves for greek bridges: methodology and case studies. Bull Earthquake Eng 7, 439–468 (2009). https://doi.org/10.1007/s10518-008-9077-2
- L109 the authors mention “the damage image”. This term is not clear.
Answer:
Means the location and severity of damage on the RC bridge.
An explanation is added in L120.
- As explained at L133, the authors model the response of the bridge in longitudinal direction as a SDoF system. Other studies adopting this approach should be mentioned (e.g. https://doi.org/10.1007/s10518-023-01783-y).
Answer:
Τhis reference is added in Introduction (L.70, see also 2 above) and in Methodology in L.147.
[25] Nettis, A., Raffaele, D. & Uva, G. Seismic risk-informed prioritisation of multi-span RC girder bridges considering knowledge-based uncertainty. Bull Earthquake Eng (2023). https://doi.org/10.1007/s10518-023-01783-y
- L305 the authors mention results of an “extended parametric analysis” which is carried out to retrieve relations between the effective stiffness and the chord rotation. Further information are required for the reader to understand the applicability of the proposed relations.
Answer:
In lines 314-360, the proposed effective stiffness scenario is detailed. New information is added to the text. This info is detailed below:
The proposed effective stiffness scenario for the cyclic-section columns of the bridge piers is established through a parametric analysis. In this analysis, various piers of different cyclic sections (D=0.6m to D=2m) and with geometric longitudinal reinforcement ratios ranging from 1% to 4% were investigated by means of consecutive pushover analyses with gradually increasing target displacements, ranging from the health state to the NC state. The analyses incorporated suitable values of effective stiffness determined through a trial-and-error process. The objective was to achieve convergence, aligning the observed chord rotation at the base section of the column piers (performance level) with the assigned percentages of reduction of the moment of inertia.
The effective moment of inertia ratios for the column piers were scaled to align with both the developed chord rotation at each target displacement (performance level) and the percentage reduction of the effective moment of inertia due to cracking and plasticization. This scaling remains within two limits: at the NC state, the effective moment of inertia ratio of an RC column pier is calculated using Eq. 7 of EN 1998-3, while just before DL, at 1st hinge formation (1st yield), the effective moment of inertia ratio of a column pier is practically equal to the 50% reduction rate proposed by EN 1998-1 for the design of new RC buildings. Similarly, the effective moment of inertia corresponding to achieving the SD performance level must align with the chord rotation at the base of the columns, which, as per EN 1998-3, is approximately 75% of that at the NC level. From the health state to the formation of the 1st hinge, the scaling approach for reducing the effective moment of inertia is constrained within the limits of the geometric moment of inertia and the 50% reduction rate proposed by EN 1998-1, respectively. A comparable scaling procedure is outlined in reference [28] for reinforced concrete buildings. During the successive pushover analyses for various column piers with different cyclic sections, the chord rotation at the base section of the column piers (performance level) converged with the assigned percentages of reduction of the moment of inertia within the analysis through a trial-and-error approach.
This approach involves plotting a curve for each pier with a distinct cyclic cross-section in a diagram to depict the effective moment of inertia ratio from the health state to the NC state. Subsequently, a mean line is drawn, enabling the utilization of the diagram for various cyclic sections of the column piers. The equations of the three lines in this diagram, representing the transitions from the health state to the first hinge, from the first hinge to DL, and from DL to NC, are also provided.
- L307 “cyclic section”. Please clarify.
Means RC piers with columns of cyclic cross-section.
Corrected in text, in L.319
- The stiffness (and the frequency) of a seismically damaged structure depends not only on the seismic deck displacement, but also on the cyclic degradation which can be related to the duration and number of cycles of the experienced ground excitation. In other words, for the same seismic target displacement, the “damaged” stiffness (and the associated modal frequency) will depend on the number of cycles experienced. The higher the number of cycles in the non-linear range, the greater the stiffness degradation and the greater the period increase, even if the maximum deck displacement is constant. Therefore, considering only the deck displacement, the proposed method ca lead to erroneous estimation. The authors should comment on this point.
Answer:
We agree with the reviewer. We note this issue in lines 649-653. We added more information according to your comment.
However, the M and P methodology utilizes nonlinear static (pushover) analysis for the bridge, a method proposed as the fundamental nonlinear analysis approach by all regulations. To clarify, considerations such as vibration duration and the number of cycles, which are dynamic in nature, are aspects addressed in dynamic (response history) nonlinear analysis. Nevertheless, in this work, we employ nonlinear static analysis as a means of identifying damage with the M and P method, while the critical parameter in the current proposed methodology for bridges is the eigenfrequency identified by the monitoring procedure under calm conditions. This frequency depends mainly on damage existence (not on the duration and the number of cycles of the seismic vibration).
- Figure 6: correct the text in the rhombus.
Answer:
Corrected in the flowchart depicted in Fig.6
- The authors state that appropriate values of the effective stiffness should be assigned to the numerical model. However, the use of a distributed plasticity model (several plastic hinges discretised along the pier height) can be useful to avoid the effective stiffness estimations. Comments on this point are required
Answer:
This approach provides an alternative method for modeling inelastic behavior, particularly suitable for purely steel structures where cross-sections do not experience cracking.
However, in reinforced concrete (RC) structures, the preferred nonlinear modeling technique involves using the effective moment of inertia of the cross-section, along with inserting a point fiber hinge at the section experiencing the higher bending moment. The length of this hinge is set equal to the plastic hinge length specified in EN 1998-3. This approach consistently aligns with experimental data for RC elements, thereby helping to avoid significant errors. The current work adopts the nonlinear modeling technique described above.
Reviewer 3 Report
Comments and Suggestions for Authors
This study introduces an intriguing approach to Damage Identification in Reinforced Concrete Bridges, employing the 'M and P' hybrid technique. This method combines a sequence of pushover analyses and immediate modal assessments, progressively targeting increased deck displacement longitudinally along the bridge. The efficacy of the suggested method is assessed using the example of a four-span bridge, specifically focusing on multi-span bridges with varying heights of piers. Before the acceptance, a few comments need to be addressed,
In Line 139, it is necessary to provide a satisfactory rationale for the omission of bridge damping, as seen in Equation 1 and subsequent equations.
In Line 147, what is the highest degree of stiffness alteration implemented in this study?
In Line 190, the definition of M and P, is suggested to be given in abstract as well.
The introductory section could be enhanced by incorporating recent studies and publications on bridge damage identification, for example: DOI: 10.1016/j.ymssp.2023.110471, DOI: 10.1016/j.ymssp.2020.107599, DOI: 10.1016/j.ymssp.2011.05.016.
The “Near Collapse (NC)” in Line 245, where the abbreviation “NC” is suggested to be given in Line 219.
Author Response
Initially we want to thank all the honorable Reviewers for their contribution to our paper.
Detailed responses are presented below to all reviewers’ comments.
Two versions of the revised paper have been prepared. The first version includes all the corrections made in the manuscript, with the originally highlighted erased text and the new corrected or inserted text presented in red font. The second version includes only the corrected text and the newly inserted text, both highlighted in red font.
It is noted that, due to the insertion of new text, particularly in the Introduction, Methodology, and References sections, the line numbering no longer aligns with the initially submitted manuscript.
Answers to Reviewers
Reviewer 3
This study introduces an intriguing approach to Damage Identification in Reinforced Concrete Bridges, employing the 'M and P' hybrid technique. This method combines a sequence of pushover analyses and immediate modal assessments, progressively targeting increased deck displacement longitudinally along the bridge. The efficacy of the suggested method is assessed using the example of a four-span bridge, specifically focusing on multi-span bridges with varying heights of piers. Before the acceptance, a few comments need to be addressed,
In Line 139, it is necessary to provide a satisfactory rationale for the omission of bridge damping, as seen in Equation 1 and subsequent equations.
Answer:
In Line 151, Eq.1 of motion (without damping) introduces the eigenvalue problem, which is addressed in Methodology. In general, the modal problem is initially solved without damping, and subsequently, damping is incorporated into the acceleration response spectra as specified in seismic codes.
In Line 147, what is the highest degree of stiffness alteration implemented in this study?
Answer:
As regards Eq. (2) in Line 159, in this work columns of pier 2 demonstrate a percentage deviation term Δki/ko of about 70% (Table 5). The mean bridge stiffness deviation relative to the health state is about 60% at a seismic target deck displacement of 8 cm along the longitudinal direction of the bridge (about after plastic hinge formation (1st yield) at the base of piers 1, 2 and 5 - see Fig.15).
In works [30] and [31], focusing on planar frame reinforced concrete structures, percentage deviation terms Δki/ko up to 90% are observed in the main diagonal of the damage stiffness matrix for a 3-story and a 5-story frame, at a seismic target top displacement which drives the structure to achieve about the SD performance level.
In Line 190, the definition of M and P, is suggested to be given in abstract as well.
Answer:
We agree with the reviewer. The definition is added in Abstract (L.10).
The introductory section could be enhanced by incorporating recent studies and publications on bridge damage identification, for example: DOI: 10.1016/j.ymssp.2023.110471, DOI: 10.1016/j.ymssp.2020.107599, DOI: 10.1016/j.ymssp.2011.05.016.
Answer:
We agree with the reviewer. These references are added in Introduction in Lines 76 to 81.
[27] Ana Fernandez-Navamuel, David Pardo, Filipe Magalhães, Diego Zamora-Sánchez, Ángel J. Omella, David Garcia-Sanchez, Bridge damage identification under varying environmental and operational conditions combining Deep Learning and numerical simulations,Mechanical Systems and Signal Processing, Volume 200, 2023,110471, ISSN 0888-3270,https://doi.org/10.1016/j.ymssp.2023.110471.
[28] Kun Feng, Arturo González, Miguel Casero, A kNN algorithm for locating and quantifying stiffness loss in a bridge from the forced vibration due to a truck crossing at low speed, Mechanical Systems and Signal Processing, Volume 154, 2021, 107599, ISSN 0888-3270, https://doi.org/10.1016/j.ymssp.2020.107599.
[29] Michele Dilena, Antonino Morassi, Marina Perin, Dynamic identification of a reinforced concrete damaged bridge, Mechanical Systems and Signal Processing, Volume 25, Issue 8, 2011, Pages 2990-3009, ISSN 0888-3270, https://doi.org/10.1016/j.ymssp.2011.05.016.
The “Near Collapse (NC)” in Line 245, where the abbreviation “NC” is suggested to be given in Line 219.
Answer:
L.219 is at the caption of Fig.1. The term “Near Collapse” is added in caption of Fig.1, now in L.232.
Round 2
Reviewer 2 Report
Comments and Suggestions for Authors
Although the evidenced limitations remain, the authors appropriately addressed this reviewer's comment.
Reviewer 3 Report
Comments and Suggestions for Authors
All comments have been addressed by the authors. The manuscript can be accepted in its current form.